# Visual evoked feedforward–feedback traveling waves organize neural activity across the cortical hierarchy in mice

Adeeti Aggarwal[1,2], Connor Brennan[1,2], Jennifer Luo[3], Helen Chung[4], Diego Contreras[1], Max B. Kelz[2] & Alex Proekt [2]✉

Sensory processing is distributed among many brain regions that interact via feedforward and feedback signaling. Neuronal oscillations have been shown to mediate intercortical feedforward and feedback interactions. Yet, the macroscopic structure of the multitude of such oscillations remains unclear. Here, we show that simple visual stimuli reliably evoke two traveling waves with spatial wavelengths that cover much of the cerebral hemisphere in awake mice. 30-50 Hz feedforward waves arise in primary visual cortex (V1) and propagate rostrally, while 3-6 Hz feedback waves originate in the association cortex and flow caudally. The phase of the feedback wave modulates the amplitude of the feedforward wave and synchronizes firing between V1 and parietal cortex. Altogether, these results provide direct experimental evidence that visual evoked traveling waves percolate through the cerebral cortex and coordinate neuronal activity across broadly distributed networks mediating visual processing.

Feedforward and feedback signaling contribute to the hierarchical processing of sensory stimuli, creating predictions and attaching behavioral context to the sensory world[1–7]. Feedforward processing involves bottom-up assembly of abstract stimulus representations in higher-order areas from simple receptive fields in the primary cortex[7,8]. Feedback processing, in contrast, involves top-down influences such as attention, prediction, and context[7,8]. Formulating predictions about the next sensory stimulus or deciding which stimulus to pay attention to requires temporal integration[1,9–11]. Thus, it is thought that feedback modulation evolves on a slower time scale relative to feedforward processing[1,2,12].

Feedforward–feedback interactions between the different cortical regions involved in sensory processing must be coordinated to give rise to integrated percepts situated in the behavioral context. The role of neuronal oscillations in coordinating neuronal activity has been a subject of intense investigation, especially in primate vision. By analyzing individual pairwise interactions between neural oscillations present at different areas of the primate cortex, many prominent

studies have shown that feedforward processing involves gamma oscillations, whereas feedback signaling uses alpha (8–12 Hz) oscillations[1,2,4,13,14]. Thus, consistent with their presumed behavioral roles, feedback signaling utilizes slower temporal oscillations compared to feedforward channels.

Pairwise interactions between oscillations in different cortical sites during processing of sensory stimuli raise several fundamental questions. Do pairwise feedforward and feedback interactions give rise to a single coherent assembly that coordinates activity among the different cortical regions involved in processing sensory stimuli? How does the brain coordinate the feedforward and feedback processing given the significant differences in timescales? One possibility for a neurophysiological process that could coordinate activity amongst multiple regions in the processing hierarchy is a spatiotemporal traveling wave. Early EEG work identified traveling waves in the feedforward and feedback directions[15–19]. However, due to the low spatial resolution of the EEG, the interpretation of these findings is unclear. Indeed, traveling waves recorded directly from the cortical surface

[1]Department of Neuroscience, Perelman School of Medicine, University of Pennsylvania, Philadelphia, PA, USA. [2]Department of Anesthesiology and Critical Care, Perelman School of Medicine, University of Pennsylvania, Philadelphia, PA, USA. [3]School of Engineering and Applied Science, University of Pennsylvania, Philadelphia, PA, USA. [4]The College of Arts & Sciences, University of Pennsylvania, Philadelphia, PA, USA. ✉e-mail: proekt@gmail.com

have different speeds and propagation patterns compared to their EEG counterparts[20–23]. Both spontaneous and stimulus-evoked traveling wave-like phenomena have been identified using voltage sensitive dyes and neurophysiological recordings from brain parenchyma in the primary and higher-order visual areas[24–34]. Most studies in this line of work, however, focused on a single cortical area rather than inter-area communication. Some studies attempted to identify spatiotemporal waves that span multiple cortical sites and concluded that sensory stimuli trigger two independent cortical waves, which travel along the horizontal fiber network in each site[25,27]. Other studies identified a reflective boundary between the primary and the secondary visual cortex[24]. Thus, while a single spatiotemporal wave of activity offers an attractive possibility for coordinating cortical activity, the existence of stimulus-evoked traveling waves with sufficient spatial scale to span the cortical hierarchy has never been directly demonstrated. Furthermore, the relationship between feedforward–feedback processing of sensory stimuli and the traveling waves evoked by them in the cortex has not been clarified.

We deploy a combination of high-density neurophysiological recordings and analytic techniques to identify large-scale spatiotemporal patterns of neuronal activity evoked by a single presentation of a simple, supra-threshold visual stimulus. By focusing our analyses on global activity patterns, rather than pairwise interactions, we show that both the feedforward and feedback aspects of visual-evoked activity form traveling waves that percolate through much of the cortex in awake mice. Feedforward waves have a fast (30–50 Hz) temporal frequency and propagate from V1 rostrally. Feedback waves are characterized by a slow (3–6 Hz) oscillation, thought to be a rodent analog of the primate alpha oscillation. These feedback waves propagate caudally from association cortices towards V1. The phase of the feedback wave modulates the amplitude of the feedforward wave, thereby forming a single multiplexed visual-evoked spatiotemporal response. Finally, we demonstrate that the feedback wave entrains firing of individual neurons in both V1 and in parietal association cortex. As a consequence, following stimulus presentation, previously uncorrelated firing in V1 and parietal cortex phase lock their firing in relation to the stimulus to form a transient neuronal assembly. Thus, we provide direct evidence that feedforward–feedback interactions organize into large-scale traveling waves evoked by simple visual stimuli. These waves serve as a scaffold that coordinates neural firing across distant cortical areas.

## Results
Our primary goal is to experimentally define salient spatiotemporal signatures of responses to simple visual stimuli. To accomplish this, we performed high-density in vivo electrophysiological recordings in awake head fixed mice ($n = 13$) (Methods for verification of wakeful states, Supplementary Fig. 1, Supplementary Movie 1). Local field potentials (LFPs) were recorded from the dural surface using a 64 channel electrocorticography (ECoG) grid placed over the left hemisphere (Fig. 1a). Two 32 channel laminar probes were also inserted perpendicular to the cortical surface targeting the primary visual cortex (V1) and the posterior parietal area (PPA). Histological and neurophysiological (Methods) localizations of the laminar probes were used to triangulate the stereotaxic locations of the individual ECoG electrodes. The ECoG grid covered a significant fraction of the cerebral hemisphere including visual, association, retrosplenial, somatosensory, and motor/frontal areas (Fig. 1b).

### Simple, brief visual stimuli evoke widespread time-locked coherent oscillations at both high (30–50 Hz) and low (3–6 Hz) frequencies
As in previous work[35–38], the visual-evoked potential (VEP) in V1 (Methods) varies from trial-to-trial. Nevertheless, early fast oscillations (30–50 Hz) followed by longer lasting slow oscillations (3–6 Hz) are

reliably identified (Fig. 1c, e, Supplementary Fig. 2). To make sure that our specific choice of filtering approach (wavelets) does not distort the data, we compared the results obtained with the wavelet-based filtering to those obtained with the conventional Butterworth filter (Supplementary Fig. 3). Analysis of the intertrial phase coherence (ITPC) confirms that these two oscillations are consistently phase locked to the stimulus (Stouffer's $p$ values < 0.00001 compared to time shuffled data) (Fig. 1d, f). ITPC computed over the first 100 ms after the stimulus reveals two peaks centered at 3–6 Hz and 30–50 Hz. The 3–6 Hz oscillation remains coherent for 500 ms after the stimulus. Because of time-frequency uncertainty, wavelet-derived ITPC (Fig. 1d, f) exhibits an artifactual increase in coherence prior to the stimulus. This artifact is eliminated when ITPC is estimated in non-overlapping temporal windows (Supplementary Fig. 4). Because these two oscillations (fast, 30–50 Hz and slow, 3–6 Hz) are reliably phase locked to the stimulus in all mice, we focus our subsequent analyses on these oscillations.

Phase locking of fast and slow oscillations to the stimulus is not limited to V1. Both fast and slow oscillations are phase locked to the stimulus across much of the cortical surface (Fig. 1g, h, respectively). Phase locking to the stimulus over large areas of the cortical surface strongly suggests that oscillations recorded at different sites are interdependent. Consistent with this suggestion, LFPs filtered at fast and slow frequencies in V1 and PPA exhibit phase coupling (Fig. 1i, j). Interestingly, oscillations at both temporal frequencies have a non-zero phase lag between V1 and PPA, as shown in Fig. 1k–l. This raises the possibility that the stimulus evokes spatiotemporal waves that percolate across the cortex. The spatial characteristics of this wave, however, are not readily apparent from just observing pairwise phase relationships. Thus, we examined the spatial characteristics of the visual-evoked oscillations that are simultaneously recorded across multiple locations on the cortical surface. Trial average LFP filtered at fast and slow frequencies along the anterior-posterior (AP) axis recorded in a single representative mouse are shown in Fig. 2a Supplementary Movies 3, 2b respectively, (see Supplementary Movies 2, 3 for propagation of visual-evoked fast and slow waves, respectively, over the cortical surface). Oscillations observed at each electrode are consistently phase shifted in relation to oscillations at neighboring electrodes. Thus, the overall ensemble activity profile resembles traveling waves at both frequency bands. Remarkably, the fast wave is initiated in the visual cortex and propagates anteriorly, while the slow oscillation initiates rostral to V1 and spreads in the opposite direction (Fig. 2c, Supplementary Movie 4). Note that traveling waves that propagate through uniform media have a uniform spatial phase gradient at all locations. Consequently, the phase offset ought to grow linearly with distance. This is approximately true of signals over short distances in Fig. 2. In contrast, over long distances a clear nonlinear relationship between phase offset and distance is seen. This nonlinear relationship implies that the propagation of these wave-like patterns is likely to depend on the specifics of network architecture.

### Coherent spatiotemporal waves are detected using complex SVD
While data in Fig. 2 strongly suggest a propagating wave-like phenomenon, the interpretation of these data is somewhat limited. First, the LFP is a complex mixture of spontaneous and evoked activity[39,40]. Second, trial averaging may obscure single trial behavior. Thus, to provide additional evidence that simple visual stimuli elicit traveling wave-like phenomena, we applied a methodology to separate spontaneous from evoked activity and to characterize spatiotemporal features of evoked activity on a single trial level. For this purpose, we utilized singular value decomposition (SVD) of the complex-valued analytical signals derived from bandpass filtered LFPs.

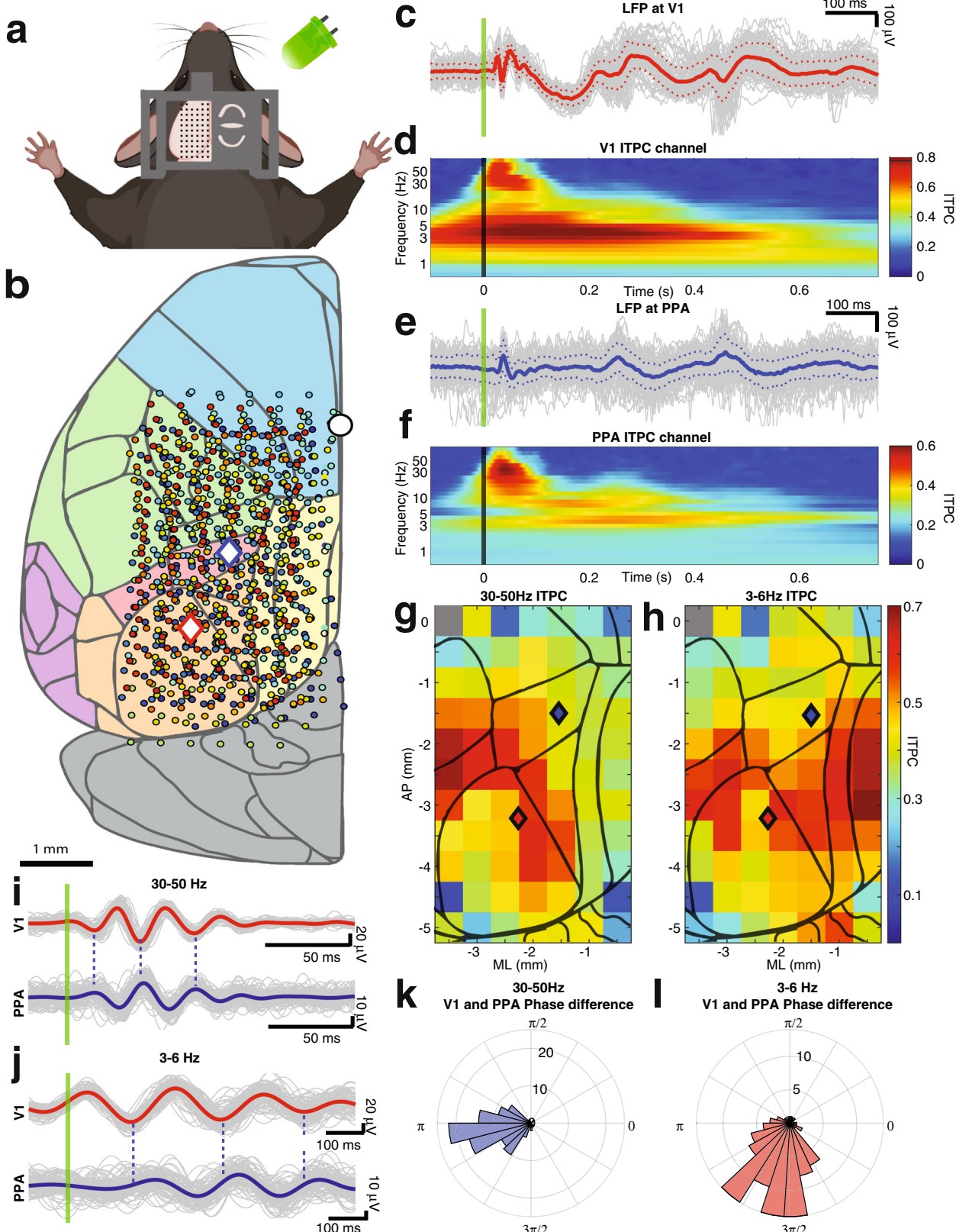

Singular value decomposition (SVD) factorizes a spatiotemporal matrix into mutually orthogonal spatiotemporal modes:

$$A = USV^T,$$

where $A$ is an $n$ by $t$ matrix that contains $n$ channels of analytical signals sampled at $t$ time points, $U$ is an $n$ by $n$ complex-valued spatial matrix in which each column encodes the phase and amplitude of a single mode at each channel, $V$ is a $t$ by $t$ complex-valued temporal matrix in which each row encodes the instantaneous phase and amplitude of each mode at each time point, and $T$ represents transposition. Finally, $S$ is an

**Fig. 1 | Visual stimuli elicit strong intertrial phase coherence over large cortical areas. a** Schematic showing the 64 channel electrocorticography (ECoG) grid used to record local field potentials (LFPs) from the cortical surface of the left hemisphere of 13 awake mice. Stimuli consisted of 10 ms flashes of a green LED placed in front of the R eye (100 trials, intertrial interval 3–4 s). Created with Biorender.com. **b** Stereotaxic coordinates of ECoG electrodes from 13 animals (color coded by animal). V1 and PPA targets for laminar probes are shown by red and blue diamond respectively. The white circle marks bregma. The cortical surface is shaded by area according to the brain regions represented in the 3D Brain Explorer of the Allen Brain Atlas[88,90,91]: visual (orange), association (red), retrosplenial (yellow), somatosensory (green), motor/frontal (blue), and cerebellum (gray). **c** Single trials, average, and standard deviation of visual-evoked potentials (VEPs) over V1 are indicated by gray, solid red, and dashed red lines respectively. Stimulus onset is denoted by the green line. **d** Intertrial phase coherence (ITPC) computed at V1 and averaged over single trials and animals (0 ms marks stimulus onset). **e** Single trials, average, and standard deviation of visual-evoked potentials (VEPs) over PPA are indicated by

gray, solid blue, and dashed blue lines respectively. Stimulus onset is denoted by the green line. **f** Intertrial phase coherence (ITPC) computed at PPA and averaged over single trials and animals (0 ms marks stimulus onset). **g** Average ITPC of 30–50 Hz oscillations within the first 100 ms of the VEP averaged over animals at each stereotaxic location. Locations in which ITPC does not meet Bonferroni corrected statistical significance compared to time shuffled surrogate data are shaded in gray. **h** Similar to E for average ITPC of 3–6 Hz activity within the first over 800 ms of the VEP. **i** Top: VEPs recorded over V1 and filtered at fast (30–50 Hz) oscillations (gray and red show single trials and trial average respectively). Bottom: Same data recorded from over PPA (gray and blue show single trials and trial average). Green line shows stimulus onset. **j** Similar to G except the signals are filtered at 3–6 Hz. Slower (3–6 Hz) oscillations also show a phase shift between V1 and the PPA. **k** Histogram of phase differences between V1 and PPA computed for 30–50 Hz oscillations. **l** Same as in **k** but computed for 3–6 Hz oscillations. *Data in **c**, **e**, **i**, **j**, **k**, **l** are from a single representative mouse.

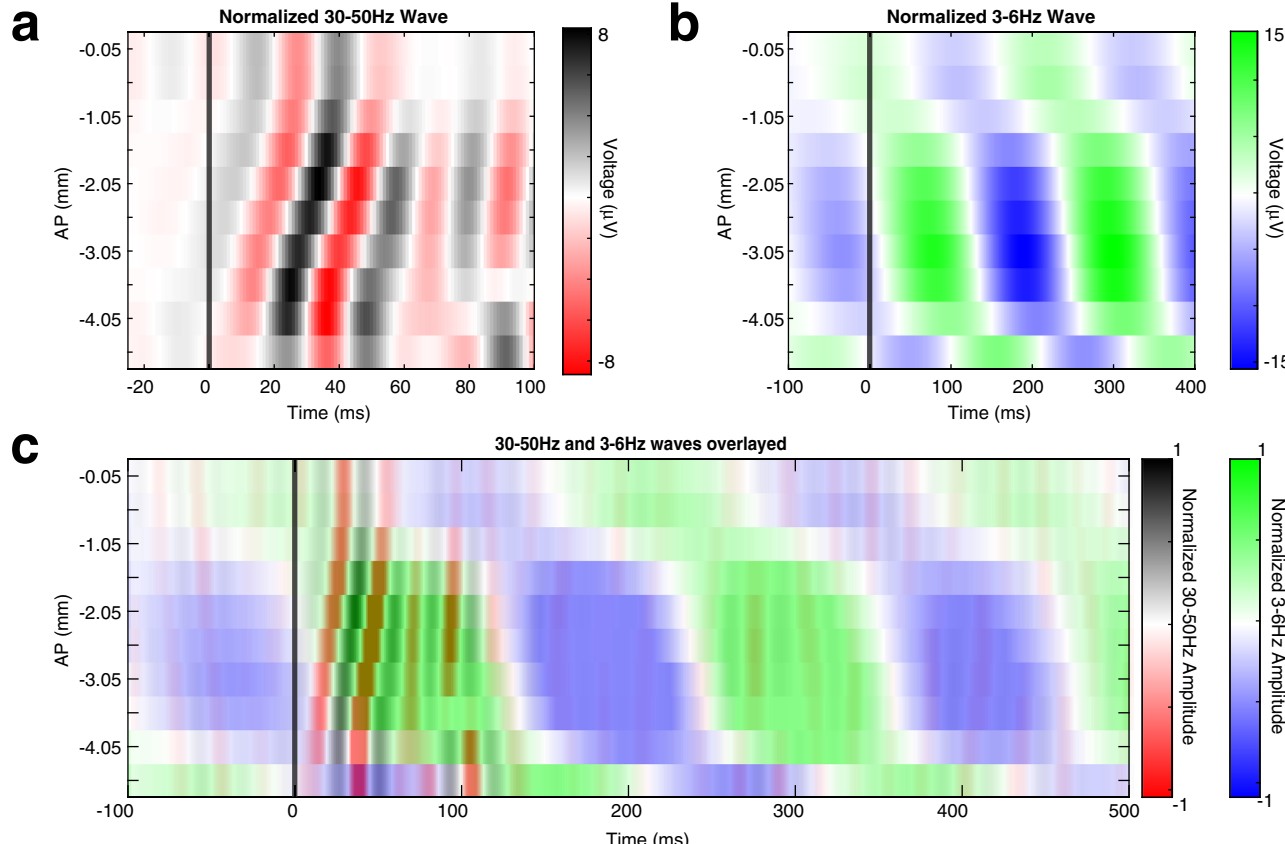

**Fig. 2 | Average filtered LFP illustrate traveling wave-like behavior. a** Average of the VEP filtered at 30–50 Hz from 10 electrodes along the anterior to posterior axis (−2.25 mm ML) in a representative mouse. The x axis denotes time relative to stimulus onset, the y axis indicates AP position of an electrode relative to bregma. Note evoked high frequency waves starting at −4.05 mm from bregma (V1) and traveling anteriorly over -100 ms. **b** Average of the VEP filtered at 3–6 Hz from the

same mouse arranged in the same format at **b**. Note the low frequency waves begin more anteriorly (-2 mm from bregma) relative to the fast oscillations and travel in the posterior direction. **c** Superimposition of the data in **a**, **b** (amplitude of the signals is normalized to highlight phase relationships between oscillations at different temporal frequencies). The fast wave begins posterior to the slow wave and travels rostrally towards the slow wave initiation zone.

$n$ by $t$ diagonal real-valued matrix which encodes the fraction of the total signal contained in each mode.

The advantage of performing SVD on the complex-valued analytical signal is that projecting the data onto the complex plane linearizes phase relationships between channels. In contrast, phase-shifted real-valued oscillations across channels would exhibit correlations at different time lags and are therefore not easily factorizable using SVD or similar dimensionality reduction techniques. We highlight the utility of complex SVD with synthetic data in Supplementary Fig. 5.

Here, we performed SVD on the analytical signal of single trials filtered at fast and slow frequencies. 72% of variance of single trial VEPs was captured by the first ten singular modes (*95% Confidence Interval* = 62–81%). We then defined the most visually responsive mode for each trial as the mode in which the post-stimulus temporal amplitude increases the most compared to pre-stimulus amplitude (Supplementary Fig. 6). The most visually responsive mode was most often associated with the largest singular value and thus contained the highest amount of signal variance. The results of the analysis are

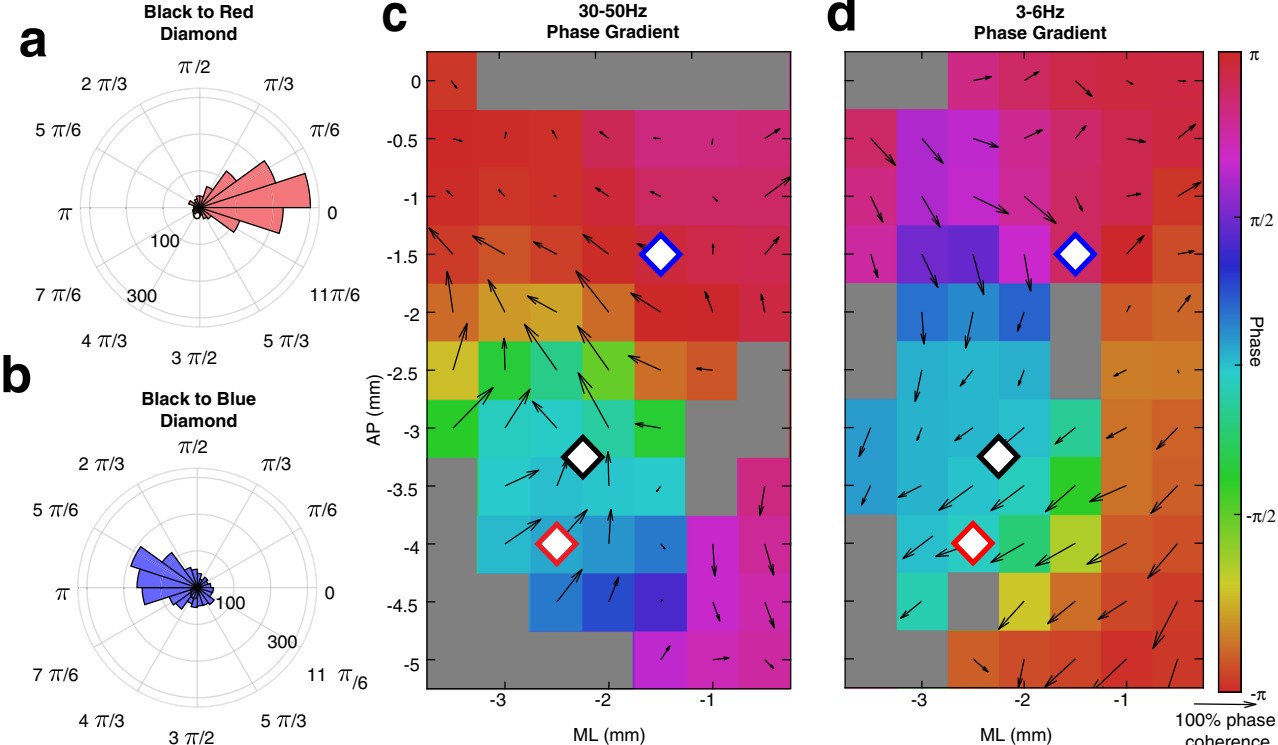

**Fig. 3 | SVD identifies visual-evoked traveling waves that are consistently elicited both across trials and across animals. a** Histogram of the difference in the spatial phase of the most visually responsive 30–50 Hz mode between two electrodes in V1 (black and red diamonds in **c**) across trials and animals. **b** Histogram of phase angle difference of the most visually responsive 30–50 Hz mode between an electrode in V1 (black diamond in plot **c**) and PPA (blue diamond in plot **c**) across trials and animals. Note that the phase angle difference is increased as distance from V1 increases. **c** At each stereotaxic location, the average phase offset of the 30–50 Hz spatial mode relative to V1 (the black diamond) is plotted in color. Spatial phase gradient is depicted by black arrows. The direction of the arrows shows the direction of spatial phase gradient over trials and mice. The length of the arrows is 1- circular variance and therefore corresponds to the consistency of the angle of the spatial phase gradient over trials and animals (scale arrow for 100% phase coherence underneath color axis in **d**). Locations that are grayed out did not meet Bonferroni corrected statistical significance (*p* value < 0.0006, Rayleigh test). **d** Phase offset relative to V1 and the spatial phase gradient of the most visually responsive 3–6 Hz spatial mode at each stereotaxic location depicted as in **d**.

robust to the changes in the total number of modes considered for the analysis.

## Visual-evoked waves have a consistent phase relationship from trial to trial and across animals

The spatial phases of the most visually responsive mode from each single trial were aggregated across trials and mice (Methods). The phase difference between visual-evoked fast waves in two locations in V1 (black and red diamonds in Fig. 3c, d) reveals a consistent phase offset across trials and mice (Fig. 3a). As the distance from the V1 electrode is increased, the phase difference between the oscillations grows concomitantly (Fig. 3b). Consistent with the average LFP data (Fig. 2), a progressive increase in phase offset with distance suggests that the activity evoked by the visual stimulus on a single trial level has characteristics resembling a traveling wave. Furthermore, the tight phase offset distribution implies that the spatial properties of these evoked waves are highly consistent from trial to trial and between animals.

The spatial phase of the most visually responsive mode averaged across trials and animals is shown in Fig. 3c, d for the fast and slow oscillations, respectively. This confirmed that throughout most of the cortical surface, the phase relationship between evoked fast and slow oscillations is consistently observed from trial to trial and among animals. Consistent with the example observed in the average filtered signal (Fig. 2), the phase gradient for fast and slow oscillations evolves in approximately opposite directions. Thus, a brief visual stimulus elicits both fast and slow spatiotemporal activity patterns that resemble traveling waves and percolate over the cortical surface for

hundreds of milliseconds. The fast wave propagates in the feedforward direction from the visual cortex towards higher-order cortical areas. The slow wave propagates in the feedback direction from the higher-order cortices back towards the primary visual cortex. Given the initiation zones and directions of propagation, we will refer to the fast visual-evoked wave as "feedforward" and the slow visual-evoked wave as "feedback." These results are robust to changes in filtering strategy (Supplementary Fig. 7).

Similar feedforward and feedback propagating waves were observed for weaker visual stimuli (Methods). For weaker stimuli, the propagation of the fast visual-evoked waves was predominantly limited to V1 and was not affected by the stimulus intensity. In contrast, the spatial extent of the feedback slow visual-evoked wave strongly depended on stimulus intensity. For lowest luminance stimulus, the feedback slow wave was principally observed in V1. However, for higher luminance stimuli, the feedback traveling slow wave involved much of the cortex (Supplementary Fig. 8). Thus, the spatial extent of the feedback wave tracks stimulus intensity, in a manner that mirrors psychophysics[41,42]. A qualitatively similar pattern for the feedforward and the feedback wave was observed after presenting static full contrast spatial gratings at two different orientations (Supplementary Fig. 9).

## High and low frequency waves are present throughout the cortical layers in V1 but are constrained to the superficial layers of the posterior parietal cortex

To identify the circuits mediating the visual-evoked waves, we computed current source density (CSD) from the two laminar probes

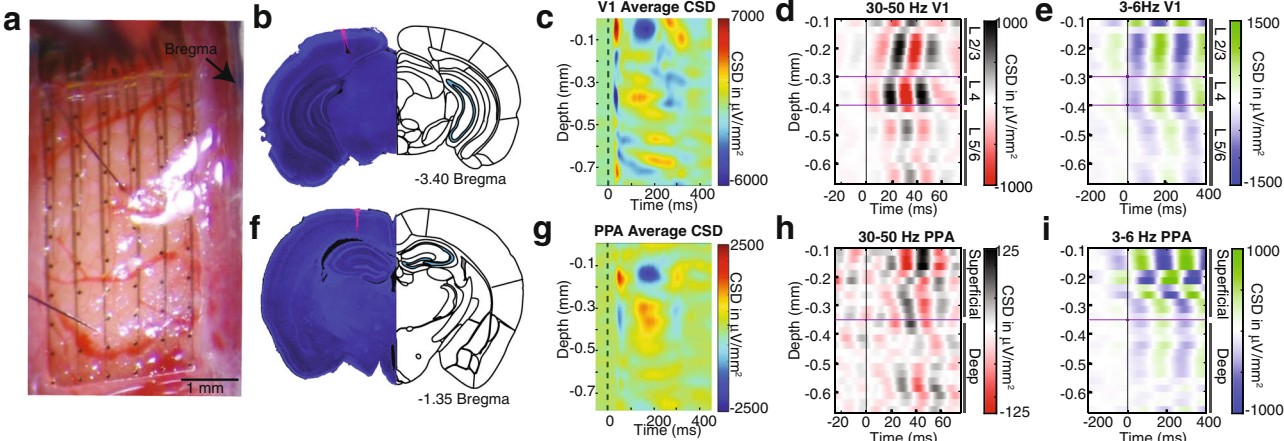

**Fig. 4 | Intra-laminar recordings reveal vertical propagation pattern of waves.**
**a** Photograph of 64 channel electrocorticography grid with two 32 channel penetrating laminar probes placed in through holes in the grid into V1 and PPA.
**b** Histological verification of laminar electrode localization in V1. Outline adapted from the Allen Mouse Brain Atlas and Allen Reference Atlas—Mouse Brain[88,90,91].
**c** Current source density (CSD) in V1 averaged over trials and mice. **d** V1 CSD filtered at 30–50 Hz and averaged across trials in a representative mouse. 30–50 Hz oscillations originate in layer IV and then propagate to superficial and deep cortical layers. In **d**, **e** purple lines show approximate location of layer IV defined by the earliest sink in the CSD. **e** Same as **d** but filtered at 3–6 Hz. The 3–6 Hz oscillations appear approximately simultaneously in the supra- and granular layers and propagate to deeper layers. **f** Histological verification of laminar electrode localization in PPA. Outline adapted from the Allen Mouse Brain Atlas and Allen Reference Atlas—Mouse Brain[88,90,91]. **g** CSD in PPA averaged over trials and mice. Purple lines show approximate location of superficial layers I–IV, and deep layers V/VI based on anatomy (Allen Brain Atlas). **h** Same as **d** for PPA. The 30–50 Hz oscillations are most prominent in the superficial layers. **i** Same as **e** for PPA. The 3–6 Hz oscillations begin in the superficial layers and propagate to deeper layers. *Data in **d**, **e**, **h**, **i** are from the same representative mouse.

targeting V1 and PPA (Fig. 4a, b, f). In V1, we identified a canonical CSD pattern. The first sink occurs in the granular layer. Subsequently, alternating sink and source patterns occur throughout the cortical column, revealing communication among the cortical layers (Fig. 4c). Less is known about the neurophysiological responses of the PPA to visual stimuli. We find the first sink at 0.15 mm below the cortical surface, which appears at a longer latency than in V1. Moreover, the majority of the CSD signal in the PPA is confined to the superficial layers (Fig. 4g).

Frequency domain analysis reveals strong ITPC for the fast frequency at all cortical layers in the first 100 ms following the stimulus in V1. A similar pattern is observed for the slow oscillation for ~500 ms after the stimulus (Supplementary Fig. 10a, c). Within the PPA, in contrast, most of the ITPC at both high and low frequencies is concentrated in the superficial cortical layers (Supplementary Fig. 10b, d). Qualitatively similar laminar profiles of ITPC at both fast and slow frequencies in V1 and PPA were observed after weaker stimuli (Supplementary Figs. 11, 12). The ITPC in both frequency bands increases with stimulus intensity.

To determine the laminar organization of fast and slow waves, we averaged the filtered CSD data at each depth within each mouse. Consistent with other work on visual-evoked gamma oscillations in V1, the fast waves originate in layer 4 in V1 and propagate to supra- and infragranular layers (Fig. 4d), indicating a critical role of thalamocortical circuitry in the initiation of the visual-evoked gamma oscillations[43,44]. In contrast, in the PPA, visual-evoked fast oscillations are predominantly seen in the superficial layers (Fig. 4h). The visual-evoked slow oscillations originate in the superficial layers in both V1 and PPA (Fig. 4e, i). These observations imply that the fast visual-evoked waves are initiated through the interactions between the thalamus and the input layer 4 of V1 and subsequently propagate through the cortico-cortical circuitry involving supra- and infra- granular layers in the feedforward direction toward higher-order cortices[45–50]. In contrast, the slow visual-evoked wave predominantly propagates in the ventral direction through the cortical column, supporting the conclusion that it is primarily mediated by the feedback cortico-cortical interactions.

## Both fast and slow visual-evoked waves have large spatial wavelengths

It is commonly thought that waves with higher frequency tend to be localized in space, whereas slow temporal frequency waves involve large areas of the cortex[51]. In contrast to these observations, we show that both the fast and the slow oscillations involve much of the cerebral hemisphere for supra-threshold stimuli. Further, examination of the recordings in Fig. 2a, b suggests that despite their difference in temporal frequency, the spatial wavelengths of both waves are similar. We confirm this observation and estimate the most common spatial wavelengths of the fast and slow waves to be 12.7 mm/cycle and 12.5 mm/cycle, respectively (Fig. 5a). These spatial wavelengths are on or above the scale of a mouse cerebral hemisphere[52]. Because of the nonlinear dependence of phase offset on distance (Fig. 2) each traveling wave does not have a well-defined single spatial wavelength. Nevertheless, local estimates of spatial wavelengths can be obtained from the spatial phase gradient (Fig. 3c, d) at each cortical site (Supplementary Fig. 13a, b). Thus, while visual-evoked fast and slow waves are distinct from canonical traveling waves in uniform medium and do not have a single spatial wavelength, the spectra of spatial wavelengths for the fast and slow oscillations are comparable. The propagation velocity of the fast oscillations, consequently, is approximately an order of magnitude faster than the slow oscillation (median$_{fast}$ = 0.8 m/s, IQR$_{fast}$ = 0.5–1.58 m/s, and median$_{slow}$ = 0.11 m/s, IQR$_{slow}$ = 0.07–0.20 m/s, for the slow and the fast waves respectively). This again is consistent with data in Fig. 2. The differences in the propagation velocities suggest that the fast and slow visual-evoked waves are mediated by different circuit mechanisms.

## Fast and slow oscillations comprise a single multiplexed visual-evoked spatiotemporal response

Until this point, we have treated the high and low frequency visual-evoked waves as independent entities. Furthermore, we only considered the waves observed in the immediate aftermath of the stimulus. However, analysis of single trials in V1 reveals rhythmic waxing and waning of the amplitude of fast oscillations aligned to the phase of the slow oscillation (Fig. 6a). Similar phase-amplitude coupling is also

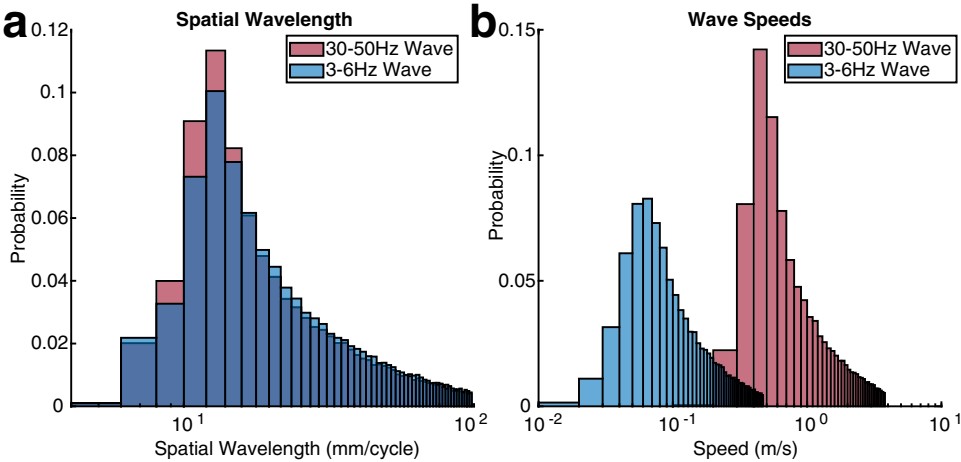

**Fig. 5 | Fast and slow visual-evoked waves have similar spatial wavelengths but significantly different propagation velocities. a** Distribution of spatial wavelengths of 30–50 and 3–6 Hz most visually responsive modes. **b** Distribution of wave speeds of most visually responsive modes at 30–50 and 3–6 Hz.

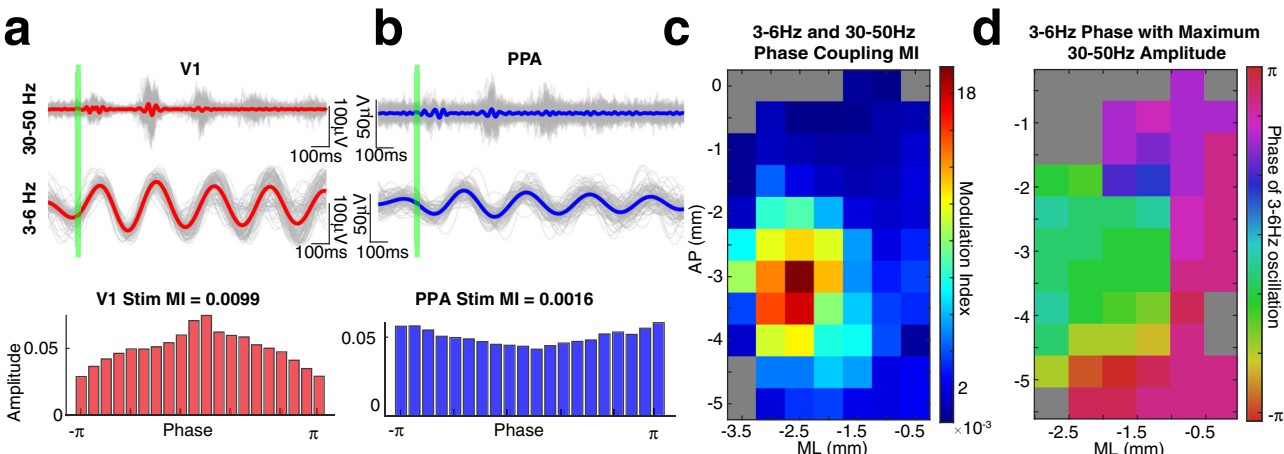

**Fig. 6 | The phase of the slow wave modulates the amplitude of the fast wave both within V1 and throughout the cortex. a** Top: single trials (gray) and average (red) data filtered at 30–50 Hz over V1 of a representative mouse. Middle: same as above, but for 3–6 Hz. Note that fast oscillation bursts occur rhythmically in phase with slow oscillations. Bottom: Amplitude of high frequency oscillations at each phase of the low frequency oscillation, averaged over trials. The deviation of this distribution from a uniform distribution is summarized in the modulation index (MI) of 0.00991 ($t$ = 36.59, $p$ value -4.9e −324, one-sided student's $t$ test, df = 99, compared to time shuffled surrogates). **b** Similar to (**a**) but for an electrode over PPA. The phase-amplitude MI = 0.00155 ($t$ = 30.24, $p$ value -4.9e−324, one-sided student's $t$ test, df = 99, compared to time shuffled surrogates). Note that the phase of the 3–6 Hz oscillation at which the gamma amplitude is maximum is shifted compared to that in V1. **c** Modulation indices averaged over all 13 mice and plotted in color at each stereotaxic location. Locations that are grayed out did not meet Bonferroni corrected statistical significance ($p$ value < 0.0006, Rayleigh test) compared to time shifted surrogate data. MI peaks near V1 but remains statistically significant over much of the cortical surface. **d** The phase of the slow 3–6 Hz oscillation at which the fast 30–50 Hz oscillation reaches maximum amplitude as shown for a representative mouse at each stereotaxic location. Grayed out locations did not meet statistical significance ($p$ value < 0.0006, Rayleigh test) compared to time shifted surrogates.

observed in the PPA (Fig. 6b); although the amplitude of fast oscillations peaks at different phases of the slow oscillation. Indeed, significant phase-amplitude modulation is present throughout the cortical surface (Fig. 6c) and peaks ~500 ms after the stimulus (Supplementary Fig. 14). Moreover, the phase relationship varies systematically with cortical location (Fig. 6d). Thus, the fast and slow waves are not independent phenomena, but instead are different aspects of the same integrated spatiotemporal activity pattern, which is reliably evoked by the visual stimulus.

### The phase of slow visual-evoked waves modulates the firing rates of neurons both in V1 and PPA

Fast oscillations in the gamma range are thought to coincide with neuronal firing. In contrast slower oscillations are dominated by synaptic potentials[53]. The phase-amplitude coupling between the fast and slow visual-evoked oscillations may therefore suggest that the slow feedback oscillation modulates neuronal firing. To determine whether this is indeed the case, we tested whether the slow visual-evoked waves entrain firing of single units in V1 and PPA. We first isolated single units throughout the cortical lamina in V1 and PPA (155 in PPA and 186 in V1, Fig. 7a, b for representative neurons in each area, respectively). Raster plots of these neurons (Fig. 7c, d) show that after the stimulus the firing of the neurons in both areas is entrained by the slow visual-evoked oscillation. To quantify this observation, we computed spike-field coherence for each single unit and the CSD filtered at the slow frequency band from the same lamina. 32 out of 155 units in PPA and 98 out of 186 units in V1 exhibited significant spike-field coherence after the stimulus (Fig. 7e, f). Spike-field coherence for the

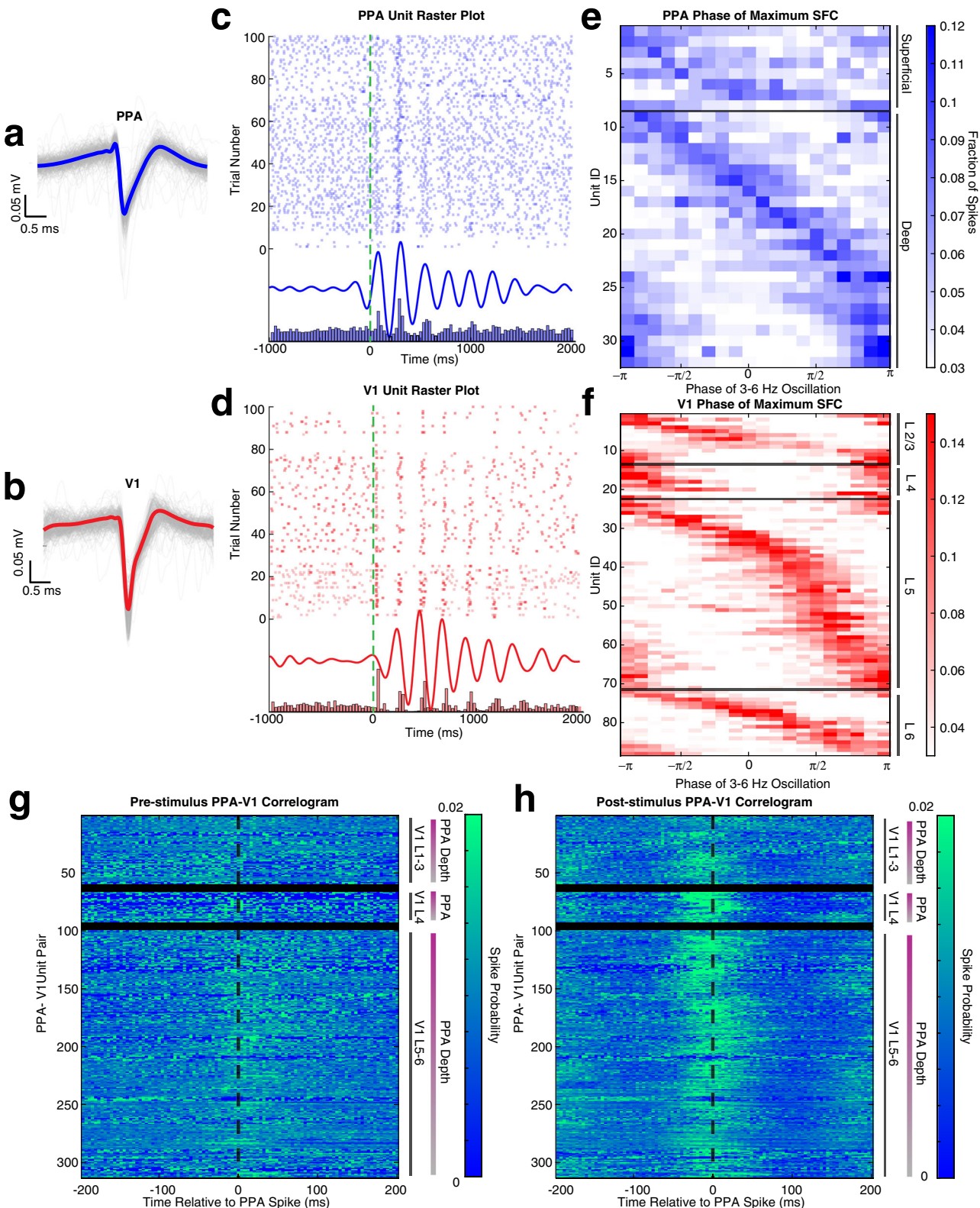

same units was significantly smaller before the stimulus ($p < 10^{-5}$ *Mann–Whitney U test*). Similar results were obtained for screen flashes (Supplementary Fig. 15). The fraction of entrained neurons increased when stimulus intensity crossed psychophysical threshold and remained approximately constant for all supra-threshold stimuli. Intersite spike-field coherence between V1 and PPA revealed significant feedforward V1$_{field}$→PPA$_{spike}$ as well as feedback interactions PPA$_{field}$→V1$_{spike}$ (Supplementary Fig. 16). Thus, visual-evoked slow

oscillations entrain a significant fraction of neurons both in the primary visual cortex and the association cortex that involve both feedforward and feedback interactions. While many single units were entrained in both cortical areas, the phase of maximum firing was not the same across different units (Fig. 7e, f). Indeed, the phase of maximum firing in each area swept through an entire cycle of the slow wave. Thus, each visual stimulus evokes a sequence of neuronal activation in both areas that is orchestrated by the slow oscillation.

**Fig. 7 | Probability of neural spiking in both V1 and PPA depend on the phase of the slow wave. a** Individual action potential waveforms of a representative PPA unit located in layer V (gray traces), with the average waveform superimposed in blue. **b** Same as **a**, but for a representative V1 neuron located in infragranular layers (gray traces), with the average waveform superimposed in red. **c** Raster plot (top) of 100 trials of PPA unit (green dashed line marks stimulus onset). The average CSD, filtered at 3–6 Hz, of the LFP at the same depth as the unit in **a** (middle). The peristimulus histogram of the same unit (bottom). **d** Same as **c**, but for the representative V1 unit in **b**. **e** Probability of firing as a function of phase of the slow oscillation for each unit in the PPA. Each row is an individual unit in the PPA that has statistically significant spike-field coherence (SFC) with the slow oscillation. Units above the black horizontal line are in the superficial layers of the PPA. Units below the black horizontal line are within the deep layers. **f** Same as **e** for V1 units with statistically significant spike-field coherence. The horizontal back lines highlight four sections in which V1 cells reside, in top-down order: layer II/II, layer IV, layer V, and layer VI. **g** Cross-correlograms between PPA and V1 neurons entrained by the slow wave during the 500 ms before visual stimulation. Each row is an individual PPA V1 pair, organized by laminar location of V1 cell and the depth of the PPA neuron from the surface (purple denotes most superficial to gray denotes deepest layers). Probability of firing is shown by color. **h** Same as in **g** but for 500 ms after the stimulus.

If the visual-evoked waves were responsible for coordinating neuronal activity across disparate regions in the cortical hierarchy, one would expect that neuronal firing would become transiently correlated after stimulus presentation. Consequently, we hypothesized that V1 and PPA neurons that are entrained by the slow wave would become transiently correlated after the stimulus. As expected, prior to the stimulus, firing in V1 and PPA was largely uncorrelated (Fig. 7g). However, after the stimulus, many of these previously independent neurons became correlated over half of the wave cycle length of the slow wave (-100 ms) (Fig. 7h). Thus, as the feedback slow visual-evoked wave propagates from the higher-order cortical areas towards the primary sensory cortex, it entrains a sequence of neuronal activation in the PPA and V1. This provides a neurophysiological insight into how simple sensory stimuli produce coordinated patterns of neuronal activity that span multiple cortical areas.

## Discussion

Here, we show that in awake mice, a brief presentation of a simple visual stimulus reliably evokes a set of two interacting traveling waves: a fast feedforward wave, and a slow feedback wave. The spatiotemporal characteristics of these waves are highly stereotyped across individual trials and across animals. Fast (30–50 Hz) waves begin in layer 4 of V1 and travel anteriorly in a feedforward manner. Slow (3–6 Hz) waves are initiated in the superficial layers of the higher-order areas and travel posteriorly in a feedback fashion. These waves are tightly coupled forming a single multiplexed spatiotemporal wave-like activity pattern observed throughout the cortex. The phase of the feedback wave modulates the firing of individual neurons both in the association cortex and in V1. A consequence of this entrainment is that following stimulus presentation, previously independent neurons in V1 and PPA form a transient coordinated assembly. In this way, the feedback and the feedforward aspects of the multiplexed visual-evoked waves coordinate neuronal activity across distant cortical regions involved in the processing of visual stimuli.

The role of neuronal oscillations in mediating feedforward and feedback sensory processing has been predominantly studied by analyzing pairwise signal covariation. Our chief contribution is that a set of such pairwise coupled neuronal oscillations together form a single coherent spatiotemporal pattern that consists of two interacting waves. Spontaneous and stimulus-evoked traveling waves have been observed in the EEG[15–19]. However, the interpretation of the EEG is hindered by low spatial resolution and volume conduction. Further, intracranial recordings (ECoG) that suffer from fewer signal distortions than the EEG, exhibited different spatiotemporal patterns compared to those discovered in EEG recordings[20–23]. Novel experimental imaging techniques using voltage sensitive dyes (VSDs) reveal mesoscopic traveling waves that are confined by anatomical boundaries between cortical regions[25,27] or produce complex interference patterns at the inter-region boundaries[24]. However, most mesoscopic waves recorded with VSDs do not take into consideration the temporal frequency of traveling waves and focus primarily on their spatial propagation properties[20,24–27,31,32,34]. Neither VSDs nor ECoG signals faithfully reflect the full richness of cortical activity. These signals are dominated by the activity in the superficial cortical layers, conflate synaptic inputs into

the surficial dendrites with their intrinsic biophysical properties, and reflect activity averaged across distinct neuronal subtypes[54]. Lastly, because the spectra of most brain signals are approximately 1/f noise, the amplitude of high frequency oscillations is orders of magnitude smaller than that of slower oscillations. Altogether these technical limitations could have contributed to a comparatively smaller extent of the fast feedforward wave detected in ECoG with weaker stimuli (Supplementary Figs. 8, 9). In contrast to the ECoG recordings, direct laminar recordings in the PPA reveal clear gamma ITPC below the cortical surface suggesting that feedforward gamma oscillations reach the PPA. Future work aimed at more complete characterization of these visual-evoked waves may need to employ a high-density 3D recording configuration.

By identifying two temporal frequencies that are reliably phase locked to the stimulus, we deconstruct the overall spatiotemporal response pattern into two distinct traveling waves that percolate through the brain in opposite directions. This, in turn, allows us to experimentally marry pairwise feedforward–feedback interactions involving different temporal frequencies and traveling cortical waves into a single, unified framework. Feedforward and feedback aspects of sensory processing serve fundamentally different roles. Feedforward processing assembles increasingly abstract representations of sensory stimuli. Feedback processing, in contrast, situates sensory stimuli within a behavioral context. Based on these functional differences, one expects that the neurophysiological processes that mediate feedback signaling must occur on a slower time scale than those involved in feedforward interactions. This assertion is consistent with experimental work in primates[1,2,4,13,14,55]. Many studies demonstrate that the faster gamma oscillations underlie feedforward processing while the slower, alpha oscillations, relay feedback processing. This difference in timescales is confirmed by our experimental observations—the temporal frequency of the feedback wave is approximately ten times slower than the feedforward wave. While there is a quantitative difference between the temporal frequency of alpha oscillations in primates and the 3–6 Hz feedback wave in our work, multiple lines of evidence strongly suggest that the 3–6 Hz wave in mice is analogous to the primate alpha oscillations[56–62]. Our identification of feedforward and feedback processes as interacting traveling waves permits an extension of this postulate. We find that the propagation velocity of the feedforward wave is also roughly an order of magnitude faster than that of the feedback wave. This slower propagation velocity of the feedback wave may contribute to the integration across multiple recent sensory stimuli.

Previous work specifically related feedback processing to perceptual modulation by attention and related phenomena. In our work, in contrast, there was no behavioral task that required or involved attention. This begs the question: What is the functional role of feedback processing in such a simple experimental paradigm? We observe that the extent of the feedback wave exhibited steplike increase for supra-threshold stimuli. The fraction of neurons entrained by the feedback wave was also increased with stimulus intensity in a similar fashion. Thus, it is tempting to hypothesize that the emergence of the large feedback wave is a neurophysiological phenomenon related to perception of the stimulus. Indeed, classic work by Verela et al.

demonstrated that long range synchronization of neuronal oscillations accompanies sensory perception[63]. These results are consistent with the "global workspace" hypothesis which proposes that consciously perceived stimuli elicit ignition-like nonlinear events that cause information about a brief stimulus to become sustained and broadcasted back through recurrent interactions between many brain areas[64,65]. Recent findings from non-human primates identified some neurophysiological counterparts of this "ignition" like event as sustained activity in the prefrontal cortex[66]. It is possible that the feedback wave identified in this study provides the scaffold that coordinates the widely distributed activity patterns that accompany sensory perception. This exciting conjecture can be tested in the future work by combining the neurophysiological techniques deployed herein with behavioral paradigms that probe sensory perception directly.

We refer to the activity patterns evoked by visual stimuli as "traveling waves". However, it is important to note that these large-scale spatiotemporal responses differ from simple waves in a uniform medium. Imagine that a response to a visual stimulus is akin to a raindrop falling into a still pond. In this highly idealized case, the raindrop would create a wave radiating outward at uniform speed and spatial wavelength. This simple scenario is indeed similar to traveling waves within a single cortical area. Much like waves on the pond, traveling cortical waves typically have tight distributions of propagation speeds and spatial wavelength[24,25,29]. Our results are in agreement with these findings over relatively short spatial scales (Supplementary Fig. 13). However, over larger scales, the apparent "viscosity" of the medium changes. There is a clear departure from the linear dependence of the spatial phase gradient on distance. This gives rise to a broad spectrum of spatial wavelengths and propagation velocities. The "viscosity" of the brain is thought to arise from conduction delays between different neuronal oscillators[52]. The observation that speed of wave propagation deviates from a pure traveling wave on large spatial scales in a systematic fashion suggests therefore that different conduction delays are involved on small and large scales. Similar phenomena have been observed in primate prefrontal cortex during working memory tasks[67]

It has been hypothesized that the interactions between distinct neuronal oscillators are mediated by horizontal fibers in superficial cortical layers. The spatial properties of traveling waves within a single cortical region are consistent with this hypothesis[25,33,68]. The propagation speeds of the visual-evoked traveling waves observed herein are also in the range of conduction delays of cortico-cortical fibers. Direct laminar recordings showing preferential involvement of superficial cortical layers provide additional evidence for this hypothesis. While on the scale of a single cortical region, wave propagation is likely predominantly mediated by horizontal cortico-cortical fibers, additional mechanisms likely contribute to propagation over large spatial scales. For instance cortico-thalamic and corticobulbar loops contribute significantly to processing of visual stimuli and involve superficial cortical layers[45–47,49,50,69–72]. We focused specifically on the interactions between V1 and PPA—an association area thought to participate in complex vision guided behaviors[73–76] and multisensory integration[77,78]. The network that includes V1 and PPA is complex and incompletely understood. In addition to the direct, spatially heterogeneous feedforward projections from V1 to PPA[79–81] and feedback projections from PPA to V1, different components of the PPA are synaptically connected to each other[80,82] and to extrastriate visual areas[69,81]. Feedforward projections from V1 to PPA commonly target layers 2–4, while feedback projections from PPA to V1 preferentially target L1 and L5[83]. Consistent with this laminar projection pattern, our intersite spike-field coherence results show that L5 V1 neurons were more likely to be entrained by the slow oscillation in PPA, while PPA neurons entrained by slow oscillation in V1 were found in layers 2, 5, 6. The relatively higher fraction of V1 neurons entrained by the PPA may indicate that the direct and indirect feedback projections from the PPA

to V1 play a dominant role in shaping neuronal firing. The interactions between V1 and PPA, however, are likely coordinated along multiple anatomic pathways.

The contribution of multiple anatomical pathways with distinct conduction velocities together with anisotropic connectivity likely distort the speed and direction of propagation of traveling waves on large spatial scales. This conjecture is supported by models of coupled oscillators which suggest that specific patterns of conduction delays strongly influence the spatial features of the wave-like phenomena[84]. These results suggest a refinement to the current mechanistic models of traveling wave phenomena. Investigation of the relationship between the underlying anatomy and propagation properties of visual-evoked waves on macroscopic scale may enrich our understanding of the relationship between neural architecture and the coordination of neuronal activity across the hierarchy of the visual system.

Our results demonstrate that the visual-evoked waves are attractive candidates for organizing the feedforward–feedback computations necessary for sensory processing. Nevertheless, the full contribution of evoked waves to the processing of visual stimuli is unclear for several reasons. First, while we note the relationship between stimulus intensity and extent of the feedback wave, we do not directly address the behavioral significance of visual-evoked traveling waves. It seems unlikely that the specific features of the waves encode sensory stimuli in a detailed way as the phase profile of the waves remained similar for spatial gratings of different orientation (Supplementary Fig. 9) and for full screen flashes (Supplementary Fig. 8). Our study specifically focused on simple unstructured stimuli to identify the dominant spatiotemporal patterns of inter-region communication evoked by them. The waves that we have identified are likely mediated by volleys of synaptic potentials[33], which modulate excitability of individual neurons. It is likely that the specific features of the stimulus are encoded by specific subpopulation of neurons that are phase locked to these waves. Processing of visual stimuli is known to involve activity of neurons broadly distributed across the visual system. Thus, while future work should address the relationship between stimulus features and the specific neuronal populations modulated by the traveling wave, our results show how such distributed neuronal assemblies can be coordinated by a wave that percolates across the visual system.

Traditional theories of sensory processing treat cortical neurons as independent feature detectors. This theoretical framework has been tremendously successful in predicting responses of individual neurons to stereotypical visual stimuli presented in isolation. However, models that treat neurons as independent feature detectors account for just a small fraction of activity in naturalistic settings[85], indicating that spatiotemporal interactions between neurons are critical for effective visual processing. For instance, laminar recordings in macaques reveal that fast feedforward and slow feedback suppressive signaling are critical for understanding the laminar dynamics of orientation tuning[86]. Recently, features of traveling waves have been associated with prioritization of neuronal responses in the new eye position after a saccade[87], detecting weak sensory stimuli[26], and cue detection during working memory tasks[67]. Our results add to this burgeoning evidence by showing that simple unstructured stimuli elicit waves of activity that percolate across space and time in a highly stereotyped fashion and entrain firing of neurons in distant cortical regions. Thus, instead of treating individual neurons as quasi–independent feature detectors, new theories of sensory processing should consider patterns of neuronal activity arising during interactions with a natural world as a superposition of waves.

## Methods
### Animals
All experiments in this study were approved by Institutional Animal Care and Use Committee at the University of Pennsylvania and were

conducted in accordance with the National Institutes of Health guidelines. All experiments were performed using 8 male and 6 female adult (12–32 weeks old, 20–30 g) C57BL/6 mice (Jackson Laboratories). Mice were housed under a reverse 12:12 h, light: dark cycle, and were provided with food and water ad libitum. A total of 20 mice were used in this study. Inclusion criteria for mice included the following: (1) presence of visual-evoked potentials (as defined by the absolute value of the average LFP response exceeding 5 standard deviations of pre-stimulus data within 100 ms after stimulus presentation), (2) histological verification of depth recording sites. With this inclusion criteria, we present data from 13 mice.

## Headplate implantation and habituation

At least 2 weeks before recording, mice were chronically implanted with custom designed headpieces for head fixation during awake recordings using standard methodology. Briefly, mice were anesthetized with 2.5% and maintained at 1.5% isoflurane in oxygen, and secured a stereotaxic frame (Narishige). Local anesthesia (0.25 ml of 0.625 mg bupivacaine) and antiseptic (Betadine) were applied. Periosteum was exposed and additional local anesthetic (0.25 ml of 2% Lidocaine gel) was applied. Bregma and lambda as well as the site of the future craniotomy (+1 mm to −5 mm AP, +0.25 mm to +6 mm ML left of bregma) were marked. The exposed skull was scored and the headpiece was attached using dental cement (Metabond) and 3 skull screws. Cyanoacrylate adhesive (Loctite 495) was applied over any remaining exposed skull. Mice were given 0.5 mg cefazolin and 0.125 mg meloxicam, and 7 ml of normal saline SQ after surgery. Animals were left to recover for a week before starting the habituation protocol. Mice were habituated to head fixation with body restraint with visual stimuli gradually over the course of 4 days. By the end of day 4, mice tolerated awake head fixation and visual stimuli for 45 min uninterrupted without any apparent distress.

## Craniotomy

On the day of the experiment, animals were anesthetized with 2.5% isoflurane in oxygen, and maintained at 1.5% isoflurane with closed loop temperature control (37 ± 0.5 degrees C) for the remainder of the surgery. 0.625 mg bupivacaine was injected in the surrounding face and neck muscles in order to provide scalp anesthesia. Mice were also given 0.5 mg cefazolin and 0.125 mg meloxicam, 0.006 mg dexamethasone and 7 ml of normal saline SQ, before surgery. Craniotomy was drilled through the dental cement over the markings on the left hemisphere (+1 mm to −5 mm AP, +0.25 mm to +6 mm ML of bregma). One of the securing screws on the right skull bone was chosen as the reference. A 64-electrode surface grid (E64-500-20-60, Neuronexus) was positioned over the dura (most medial and anterior electrode was positioned ~1 mm lateral and 1 mm posterior to bregma). Two laminar 32 channel probes (H4, Cambridge Neurotech) were coated with DiI (Sigma-Aldrich) for postmortem histological localization. The probes were inserted through the hole in the ECoG grid closest to V1 (−3.25 AP, −2.25 ML) and PPA (−1.5 AP, −1.5 ML) using a motorized micromanipulator (NewScale Technologies). Electrodes were inserted 800 μm into the brain at a rate of 25 μm/min. V1 electrode position was verified with current source density analysis. The grid and exposed dura was then covered with gel foam soaked in mineral oil. Isoflurane was then turned off for at least 20 min. At the end of this period and prior to recordings, animals were whisking, moving limbs and blinking in a manner similar to habituation before recordings began, thus suggesting that they were awake. This was corroborated by online analysis of the ECoG. After visual stimulation and recording, animals were deeply anesthetized (5% isoflurane) and sacrificed. Brains were extracted and fixed in 4% paraformaldehyde (PFA) overnight prior to sectioning and histology.

## Histology

Brains was sectioned at 80 μm on a vibratome (Leica Microsystems). Sections were mounted with medium containing a DAPI counterstain (Vector Laboratories). Electrodes were localized using epifluorescence microscopy (Olympus BX41) at ×4 magnification.

## Visual stimulation

Visual stimuli consisted of a 10 ms flash of a green LED (650 cd/m²) separated by random intertrial intervals sampled from a uniform distribution between 3 and 4 s. The flash covered 100% of the mouse's visual field. Visual stimulation was also performed using a CRT monitor (Dell M770, refresh rate 60 Hz, maximum luminance 75 cd/m², positioned 23 cm away from the mouse's right eye, at an angle of 60% from the mouse's nose, thereby covering 70% of the mouse's right field of view) at varying luminance (2%, 11% 44%, 75%, 100% of maximum screen luminance). Flashes were 100 ms long and presented in a random order at a random time interval between 3 and 5 s. In a subset of 8 animals, 280 trials of 500 ms full screen static gratings (0.4 cycles per degree, 100% contrast) oriented at either 45° or 135° from horizontal were shown on the same CRT monitor at a random intertrial interval uniformly distributed between 2 and 3 s.

## Electrode registration

After identifying the histological location of the two depth probes in each mouse, and with prior knowledge of the ECoG grid dimensions (i.e., 6 columns, 11 electrode rows, electrode spacing of 500 μm, electrode diameter of 60 μm, hole diameter of 200 μm,), the position of each ECoG electrode was triangulated in the following fashion. The ECoG grid is a semiflexible plane. The location of the cortical probes in the electrode coordinate system was given by the through holes used for electrode insertion. The stereotaxic coordinates of the electrodes were established using postmortem histology by comparison to the brain atlas[88]. The cosine of the angle, $\theta$, between the laminar probe positions in the electrode and stereotaxic coordinates was computed. Each electrode on the ECoG grid was assigned a location based on the Euclidean distance from the two laminar probe sites. The resultant grid location matrix was then multiplied to a rotation matrix (R) to obtain the final electrode positions in stereotaxic coordinates.

$$R = \begin{bmatrix} \cos\theta & -\sin\theta \\ \sin\theta & \cos\theta \end{bmatrix}$$

These coordinates were then verified by comparison to photographs of grid positions taken during experimental session.

## Electrophysiology and preprocessing

Signals were amplified and digitized on an Intan headstage (Intan, RHD2132) connected to an Omniplex acquisition system (Plexon, Omniplex), and streamed to disk at 40 KHz/channel. Impedances for the electrodes on the surface probe ranged between 0.18 and 0.35 MΩ. Impedances of the channels on the laminar probes ranged from 0.05 to 0.06 MΩ.

To extract the LFP, data were downsampled to 1KHz and filtered offline using a custom-built FIR filter between 0.1 and 325 Hz, with the MATLAB functions, *firls.m* and *filtfilt.m* to minimize phase distortion. Noise channels were manually removed and trials with excess motion artifact were rejected. Subsequently, the ECoG signals were mean re-referenced to minimize the effect of volume conduction. All further analysis was performed using custom-built Matlab (Mathworks) code unless otherwise stated.

## Selection of electrode over primary visual cortex (V1)

To average over animals, a single stereotaxic V1 location was selected as the electrode closest to (−3.25 AP, −2.25 ML), and in each animal. To confirm that the chosen electrode neurophysiologically corresponded

to V1, the latency of onset of the VEP at each grid electrode was computed. The latency of onset of the visual-evoked potential was calculated as the time point at which their post-stimulus average exceeds 3 standard deviations above the pre-stimulus baseline for 3 consecutive time points. The stereotaxically labeled V1 electrodes were 1–2 electrodes away from the electrodes with the earliest latency of onset in all mice and had latencies of onset within 2 ms of the electrodes with the earliest latency of onset.

### Current source density analysis (CSD)

The one-dimensional CSD was computed as the second spatial derivative of the LFP recorded from the linear probes (Freeman and Nicholson, 1975):

$$\frac{d^2\varphi}{dz^2} = \frac{-[\varphi(z + 2\triangle z) - 2\varphi(z) + \varphi(z - 2\triangle z)]}{(2\triangle z)^2}$$

where $\varphi$ is the LFP, z is the vertical coordinate depth of the probe, and $\Delta z$ is the interelectrode distance (25 μm). CSD at the electrode boundaries were obtained using estimation procedure in ref. 89. Cortical layers in the V1 probe were identified by the pattern of visual-evoked current sinks and sources (colored as blue and red, respectively). Channels with the earliest current sink were assigned as layer 4 (granular layer). Subsequent sinks were found above and below layer 4 in layers 2/3 and layer 5. Laminar assignment of the channels in the PPA probe were based on distance from the cortical surface, where the CSD converged to zero. Channels within the first 350 μm were defined as superficial layers based on the thickness of layers 1–4 of the PPA[88,90,91]. The next 400 μm were defined as deep layers. All further analysis of laminar LFP data was performed on the CSDs. V1 probe data was included only if there was a clear layer 4 sink and subsequent layer 2/3 and layer 5 sinks. Similarly, PPA probe data was included in analysis only if the superficial loss of CSD was seen, indicating that the most superficial electrode was positioned at the cortical surface. 11 mice fulfilled these criteria and were included in the analysis of laminar recordings.

**Spike sorting.** Single unit identification was performed on probe data from the same 11 mice that fulfilled criteria for laminar analysis. Spike sorting was performed using Kilosort[92]. The resulting spikes were then manually inspected for correct waveform clustering using Phy. All units with a firing rate lower than 0.5 spikes/s were excluded from further analysis.

### Wavelet analysis

Power, phase, and frequency information was extracted using a continuous wavelet transform using Morlet wavelets (0.1 to 150 Hz, with a step-width 0.25 Hz and normalized amplitude) (available at: http://paos.colorado.edu/research/wavelets/)[93].

### Multitaper spectral analysis

The power spectra in Supplementary Fig. 2 were generated using Thomson multitaper method (5 tapers) implemented in Chronux function *mtspectrumc.m* (found at http://chronux.org/)[94]. 150 and 1000 ms windows immediately before and after the stimulus were used to estimate the power of 30–50 and 3–6 Hz oscillations, respectively.

### Intertrial phase coherence (ITPC) analysis

Intertrial phase coherence (ITPC) was used to quantify the phase synchrony between trials at each point in the time- frequency plot. ITPC was calculated for each electrode in each mouse. Briefly, complex-valued wavelet coefficients were projected onto a unit circle by taking the wavelet phase at each time point and each frequency and setting the length of the vector to one. ITPC was then calculated as the

circular average of such vectors across trials at each point in the time-frequency plane[95].

Because of time-frequency uncertainty, the wavelet-based ITPC estimation for lower frequencies produces an artifactual increase in coherence prior to the stimulus. To better define the time course of ITPC, we additionally estimated this quantity using multitaper methods. ITPC was computed in non-overlapping 100 ms or 450 ms windows for 30–50 Hz and 3–6 Hz ITPC respectively and averaged across all surface electrodes for all animals.

### Filtering data with wavelet coefficients

LFP or CSD data was filtered into high (30–50 Hz) or low (3–6 Hz) frequency bands in order to perform phase based analysis. Data was filtered using the inverse wavelet transform, *invcwt.m*, (available at: http://paos.colorado.edu/research/wavelets/)[93], by setting all wavelet coefficients outside the desired frequency range to zero.

### Filtering data with Butterworth filter

To assure that the results are not affected by choice of filter strategy LFP was also filtered into high (30–50 Hz) or low (3–6 Hz) frequency bands using a 6th or 3th order Butterworth filter, respectively. To avoid phase distortions, the Butterworth filter was passed forward and backward across the data (implemented as *filtfilt.m* in Matlab). The comparison of wavelet filtered vs. Butterworth filtered data is shown in Supplementary Figs. 3, 7.

### Analytical signal extraction

Hilbert transform was used to derive the analytical signal of LFPs or CSDs filtered in the gamma (30–50 Hz) and low frequency (3–6 Hz) data. This produced a time series of complex numbers. The modulus of the analytical signal is the instantaneous amplitude while the instantaneous phase is given by its arctan.

### Complex singular value decomposition (SVD)

LFP recorded during a single trial and filtered at the appropriate frequency range (see above) were Hilbert transformed to derive an $n \times t$ analytical signal matrix $A$, where $n$ is the number of electrodes and $t$ is the number of time points. Spatiotemporal modes were extracted from $A$ by performing singular value decomposition which factorizes $A$ into mutually orthogonal modes:

$$A = USV^T,$$

where $T$ denotes transpose. The columns of complex-valued $U$ and $V$ are the left and right singular vectors, which encode the spatial and temporal components of each mode, respectively. The diagonal real-valued $S$ contains singular values ($\lambda's$). The fraction of the total signal explained by $i - th$ mode is given by

$$\lambda_i / \sum_1^n (\lambda),$$

The spatial amplitude of each mode is computed as $w_s = |U_{(*,i)}| * \lambda_i$. Each of the $n$ components of $w_s$ reflects the contribution of each electrode to the mode. The spatial phase is defined by $\theta_s = \arctan U_{(*,i)}$. Each component of $\theta_s$ reflects the spatial phase of each electrode. Temporal phase $\theta_t$ and amplitude $w_t$ are defined in a similar fashion from $\mathbf{V}$, $\theta_t = \arctan V_{(i,*)}$ and $w_t = |\mathbf{V}_{(i,*)}| * \lambda_i$. The spatial phase gradient of each mode $\frac{d\theta_s}{ds}$ can then be computed locally at each electrode as in ref. 25 (see below). The temporal frequency is given by the time derivative of the unwrapped temporal phase $\frac{d\theta_t}{dt}$ normalized by 2π. The analytical signal corresponding to mode $i$ can be reconstructed as $A^i = U_{(*,i)} * \lambda_i * V_{(i,*)}^T$. Finally, the LFP signals corresponding to each mode can be reconstructed as $P^i = |A^i| * \cos(\arctan(A^i))$. Illustration of this procedure is shown in Supplementary Fig. 4.

## Defining the most visually responsive mode

The first ten singular modes (accounting for 62.34–81.18% variance, 95% confidence interval) computed for each single trial, as above. $w_t$ is defined as the temporal amplitude for the first ten modes. $w_t$ during the pre-stimulus period (400 ms) and was then used to compute the mean, $\langle w_t \rangle$ and the standard deviation, $\sigma_{w_t}$. $w_t$ for the entire trial period (pre- and post-stimulus) was expressed as a z-score $w_z = (w_t - \langle w_t \rangle)/\sigma_{w_t}$. The most visually responsive mode was defined as the mode that exhibited the greatest increase in amplitude during the post-stimulus period (defined as 350 ms post-stimulus for fast oscillations and 1000 ms post-stimulus for slow oscillations) for the LED and full screen flash. For spatial gratings, the window was 350 ms for fast and 1500 ms for slow oscillations.

## Spatial phase offset from V1

To determine the consistency in the phase relationship between spatial modes identified in different trials and across animals, the average difference in phase from each channel to the V1 channel was computed for the most visually responsive spatial mode. The V1 channel in each animal is defined as the channel closest to (−3.25 AP, −2.25 ML). The phase offset from V1 is calculated at each electrode by extracting the spatial phase of the most visually responsive mode $\theta_s$ and setting the V1 phase to zero. Circular mean and variance of $\theta_s$ referenced to V1 was then computed across trials and across animals[96]. The direction of the resultant vector corresponded to the average phase, whereas the magnitude of the vector is 1-circular variance.

## Spatial phase gradient

The spatial phase gradient for the visually evoked mode was quantified as follows. For each trial at each electrode the complex-valued spatial loading was multiplied by the complex conjugate of the spatial loading of its adjacent electrode implemented in *phase_gradient_complex_multiplication.m*, (available at: https://github.com/mullerlab/wave-matlab)[25]. This operation was performed iteratively along the AP and ML direction of the grid. The resulting vectors were then converted into polar coordinates. To quantify the average gradient over trials, each trial's gradient vector at each location was projected onto a unit circle and circular average was computed. The angle of the resultant average vector is the direction of the average phase gradient, whereas its magnitude is 1-variance of the gradient over trials.

## Spatial wavelength

The spatial frequency was computed by multiplying the magnitude of the single trial spatial gradient vectors and dividing the result by $2\pi$ to convert the units into cycles/mm. Spatial wavelength was calculated as the reciprocal of spatial frequency.

## Velocity of visual-evoked waves

The instantaneous temporal frequency was computed by measuring the slope of the unwrapped temporal phase of the most visually responsive mode. This yields the temporal frequency $f_t = \frac{d\theta_t}{dt}$. The spatial gradient at each location (x,y) is a two component vector $\langle \frac{d\theta_s}{dx}, \frac{d\theta_s}{dy} \rangle_{(x,y)}$, where dx and dy are unit vectors in the mediolateral and the anterior-posterior directions respectively. The Euclidian norm of this vector is $\nabla_s = \sqrt{\left(\frac{d\theta_s}{dx}\right)^2 + \left(\frac{d\theta_s}{dy}\right)^2}$. Wave velocity at each location is then defined as $v = \frac{f_t}{\nabla_s}$.

## Phase-amplitude coupling

Phase-amplitude coupling between oscillations was assessed using the modulation index (MI) of single trial filtered LFP data at every grid electrode[97]. Phase of the 3–6 Hz filtered data and the amplitude of the 30–50 Hz filtered data were extracted from the analytical signal $A$ as described above. Phase was binned into 20 phase intervals. The mean 30–50 Hz amplitude was calculated for each bin for the first 900 ms of post-stimulus activity per trial in Fig. 6. In Supplementary Fig. 14, this calculation was broken into the first 0–450 ms interval and the second 450–900 ms interval immediately after the stimulus. The mean amplitude per bin was then averaged over trials. The MI was calculated by measuring the divergence of the resulting amplitude distribution from the uniform distribution using a modified Kullback–Leibler (KL) distance metric, with the function, *ModIndex_v2.m*, (available at https://github.com/tortlab/phase-amplitude-coupling)[97].

**Spike-field coherence (SFC).** The phase of 3–6 Hz filtered CSD was extracted from the analytical signal and segmented into 20 phase bins. The mean spike count was calculated for each phase bin for the first 900 ms of post-stimulus activity per trial, and then averaged over trials. The SFC was quantified as described above for MI.

**Intersite spike-field coherence (SFC).** To determine how the phase of the 3–6 Hz oscillation in one region affected spiking probability within a different brain area, spike-field coherence was calculated as above using the spike data from each entrained neuron in V1 and PPA and correlating the firing pattern to the phase of the 3–6 Hz oscillation at each location in PPA and V1, respectively.

## Averaging signals over stereotaxic coordinates

A query grid of stereotaxic locations was defined spanning −3.5 to 0 mm ML and −5 to 0 mm AP with 0.5 mm spacing. For each query location, the weight of each electrode was assumed to depend on the distance between the electrode and the query location. The weights were defined to be a Gaussian function of Euclidian distance from query location as follows:

$$W = \frac{1}{\sigma\sqrt{2\pi}} e^{\frac{||p_q||}{\sigma}},$$

where $||p_q||$ is the Euclidean distance between the electrode p and query location q, $\sigma = 0.15$ is the standard deviation. The weight was computed as in the above equation for all electrodes within 0.3 mm of the query location and was set to zero otherwise. This weighting was used when computing averages and variances across mice as a function of stereotaxic coordinates.

## Spike correlations

For each entrained V1 and PPA neuron in each animal, the delay of the spike times of the V1 cells relative to the spike times of PPA cells was computed during the pre-stimulus timeframe (500 ms before the stimulus onset), and post-stimulus timeframe (0–500 ms after stimulus onset). The distribution of spike time delays is displayed in Fig. 7.

## Statistical analysis

Statistical significance threshold for all measures was set to *p = 0.05*. To compute *p values* for ITPC, SFC, and MI method of shuffled surrogates was used. For ITPC, shuffled surrogates consisted of trials where the stimulus time was assigned randomly. For MI, shuffled surrogates consisted of shifting the phase time series by a random amount relative to the amplitude time series. For SFC, shuffled surrogates were given by a Poisson model constructed for each neuron. The spike rate for each neuron was computed in 200 ms time bins shifted by 1 ms. Spike times for the shuffled surrogates were then produced by simulating the Poisson process given by this spike rate. For all *p* value calculations, 100 shuffled datasets constrained to have the same number of trials as experimental data were produced. The mean and the standard deviation of the quantity of interest (ITPC, SFC, or MI) was then computed for the shuffled surrogates. Finally, the

experimentally observed quantity of interest was expressed as a *z-score* relative to the shuffled surrogates.

For establishing statistical significance for ITPC, *z-score* (as described above) was computed for each experiment at each stereotaxic location. To determine the aggregate *p* value over mice at each stereotaxic location, a Stouffer's Z-score was calculated across experiments. To establish statistical significance of spatial phases (see above), a Rayleigh test was performed using circ_rtest.m (available at http://www.jstatsoft.org/v31/i10)[98]. In all cases the threshold *p* value for statistical significance was adjusted for multiple comparisons (multiple spatial locations) using a Bonferroni correction.

To establish the statistical significance of the effect of stimulus intensity on the proportion of cells with significant SPC (defined above), Tukey's HSD multiple comparisons test was performed using the function *tmcomptest.m*[99].

### Reporting summary
Further information on research design is available in the Nature Research Reporting Summary linked to this article.

## Data availability
We added our data and code to the following repository: https://doi.org/10.5281/zenodo.6578571 Source data are provided with this paper.

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

## Acknowledgements
We wish to thank Brenna Shortal for setting up Kilosort. We also want to thank Dr. Andrew Hudson, Dr. George Mashour, Dr. Marcelo Magnasco, Dr. Joseph Cichon, Dr. Andrew McKinstry-Wu, Andrzej Z. Wasilczuk, and Ethan Blackwood for helpful discussions. This research was supported through the Translational Neuroscience Initiative from the Penn Medicine Translational Neuroscience Center (PMTNC), RO1 GM088156 (M.B.K.), RO1 GM124023 (A.P.), T32 EY007035 (A.A.), F30 EY029931-01A1 (A.A.).

## Author contributions
A.A., D.C., M.B.K., and A.P. conceptualized the study design. A.A., H.C., D.C., and A.P. curated the data. A.A., C.B., J.L., and A.P. performed formal analysis. A.A., D.C., M.B.K., and A.P. acquired funding. A.A., M.B.K., and A.P. wrote the initial draft, and A.A., C.B., J.L., M.B.K., D.C., and A.P. participated in writing, review and editing the paper.

## Competing interests
The authors declare no competing interests.
