## [Peer Review File · Nature Communications]

Visual evoked feedforward-feedback travelling waves organize neural activity across the cortical hierarchy in miceREVIEWER COMMENTS

Reviewer #1 (Remarks to the Author):

In this work, the authors study ECoG and laminar recordings from awake mice following presentation of brief visual stimuli. By analyzing evoked activity patterns at both trial-average and single-trial level, they find the evoked response has a sophisticated spatiotemporal structure, with a narrowband feedforward wave in the 30-50 Hz range and a feedback wave in the 3-6 Hz range.

The results are timely and important; however, several major conceptual and technical points need to be addressed.

MAJOR

(1) A major concern is that the visual stimulus is far from physiologically realistic. This is not necessarily a detraction; however, the authors need to seriously consider what this means for their work. In standard visual paradigms, stimuli are either carefully constructed, titrated to produce specific visual responses, or matched to natural scenes. Further, it is a real possibility that the specific gamma activity and highly aligned spiking are result of the authors' specific and different stimulus presentation. The authors need to consider seriously this stimulus in the context of their work. While a strong, transient stimulus can certainly give much information about the system, it needs to be established that this information is general to how this system functions in other conditions, which already may not be the case with weaker visual stimuli (line 212-213). Additional experiments may be needed to clearly establish this because it's a serious concern.

(2) The authors need to make a clear statement about what the feedback could be doing in this case. Top-down, feedback connections are largely thought to be anticipatory or modulatory. It's highly interesting that feedback activity could be organized as a wave; however, given the visual stimulus, it's highly unclear what feedback could be doing in this case. What is being anticipated or modulated in this experiment? This is a serious concern and one that must be addressed for this work to have impact in neuroscience. Finally, both the progression of activity across the visual system (as a feedforward "sweep") and the organization of feedforward-feedback dynamics across different frequency bands have been previously described in the work of Roelfsema. While specifically identifying these dynamics as a traveling wave is certainly interesting, the authors need to clearly describe how these results differ from that in previous work.

(3) It is important to consider potential distortions from narrowband filtering throughout this work. For both the 3-6 Hz and 30-50 Hz frequency band, it would be important to see examples of raw activity patterns, along with systematic quantification, to understand whether observed effects are due to true oscillations in each band. Importantly, activity near 50 Hz is known to contain spike waveform artifacts (Ray and Maunsell, PLoS Biology, 2008). Effects observed here in the higher frequency range extend up to about 80 Hz (cf. Fig 1d), clearly within the affected frequency range. Removing these artifacts, for example with the algorithm proposed by Zanos et al. (J Neurophys, 2011), has been observed to significantly impact spike-field relationships in this frequency range. Finally, the filtering approach of setting wavelet coefficients to zero (line 556-559) may not be optimal in this case, as it is almost reminiscent of a boxcar filter in the Fourier domain. The authors should establish that this is free from artifact and explain why this is a useful method.

MINOR

- line 21: "It has been hypothesized...": By whom? Has this specific hypothesis of traveling waves been previously addressed?

- line 332: "Further, intracranial": ECoG recordings have certainly shown different patterns from early EEG; however, it may be incorrect to say they did not corroborate this earlier work.

- line 357: "numerical discrepancy": This word choice makes the difference seem like some sort of a mistake. Perhaps "quantitative difference" would be more clear.

Reviewer #2 (Remarks to the Author):

This is a very interesting paper on visually evoked oscillatory activity in awake mice recorded using large ECoG grids (covering multiple areas in the mouse brain) providing a vast amount of novel analysis/ information on traveling waves in visual and association areas, and this within the known high frequency (gamma)/ feedforward and slow (theta) frequency/ feedback framework of signal processing. It is suggested that the demonstration of traveling waves in this context resolves an outstanding question in systems neuroscience: how the brain coordinates bidirectional network activity.

The paper is well written and the text and Figures guide well through the data. The results provide a lot of interesting information. The paper is in very good shape already. I don't have much suggestions for improvements/ additions.

Specific points (all discussion points)

1) I am convinced of the existence of traveling waves by the evidence presented. One thing that intrigues me though is how traveling waves can be reconciled with the concept of functional networks and connectivity. Functional and anatomical connectivity will in many instances skip cortical boundaries so that shortest paths for signal propagation will not always continuously follow the cortical surface, which is what traveling waves seem to largely do. Hence, there seems to be a fundamental difference between the function of signal propagation through anatomical connections versus along waves propagating across the cortical surface. I am unsure how to reconcile them. If waves travel across the cortex only when anatomical connections coincide with a more continuous connectivity pattern, then the term traveling waves I think is misleading, as whether waves are observed hinges on specific connectivity. If this is not the case, I am unsure how to integrate them in for instance signal propagation along larger fibre bundles that connect remote regions, e.g. the corpus callosum or the superior longitudinal fasciculus. Could this be discussed in more detail?

2) Another point that I find intriguing is that the slower (theta) oscillations occur much later in the trials than the faster (gamma) oscillation. How can this be reconciled with the prediction framework (referred to in the paper) where slower oscillations should appear before faster oscillations so that top-down control can occur? Could this be resolved?

Reviewer #3 (Remarks to the Author):

The current manuscript by Aggarwal A., et. al. characterizes the spatiotemporal patterns of visually evoked potentials across multiple frequency bands as travelling waves and propose such waves as a potential mechanism for feedforward-feedback visual (or potentially sensory in general) processing at the cortical level. Specifically, a large-scale simultaneous recording of cortical surface coupled with laminar recording from V1 and PPA was employed and showed that low gamma oscillation feedforward waves were initiated in V1 and propagated rostrally, while a rodent “alpha” oscillation over 3-6 Hz range feedback waves modulated the amplitude of the feedforward waves synchronizing spiking activities between V1 and PPA. The overall study design and the data are well done (except for the somewhat repeated stimuli that appeared to evoke responses prior to the stimulus onset), some of the analyses were not direct to show what the authors wanted to show. Furthermore, although at the very high level there was an intellectual motivation for the study, authors should include more citations for anatomical and physiological studies as opposed to cite more computationally driven studies. For the detailed comments, please see below.

Line 72: Most of this work -... > Most in this line of work ...

Line 118: In Figure 1c, it is not clear how variable across trials the responses are during ~ 500-600 or 650 ms after the stim onset, and how the average peak is much smaller than the range of responses shown in individual gray traces... Not just the mean to be illustrated but adding std around the mean can be informative.

Lines 120-1: First of all, what is the statistical threshold for the significant ITPC based on the scale color-coded on the right of Figure 1d? Second, the ITPC values across 3-10Hz before the visual cue onset appear to be similar or in the same range as the “early” phase for the low gamma range. What is the logic here to say (or not to say) anything about the pre-stimulus response? Is this response a result of the semi-rhythmic stimuli (3-4 sec) so that animals could predict roughly when the stimulus appears? Also in other figures, V1 and PPA responses are compared, so why not showing the response of PPA channel in the same way as in Figure 1d?

Lines 126-132: The early response, mentioned above in relation to Figure 1d, is prominent and visible in Figure 1i where V1 signals clearly have non-flat response compared to all the other three signals depicted in Figure 1 h and i. Also, it is not clear how readers are supposed to be convinced to have clear phase coupling from Figure 1i. Other measures such as cosine similarity or circular time series correlations to show statistically significant phase coupling in all the cases, especially in Figure 1i for the slower oscillation.

Lines 133-149: The authors’ intent is somewhat visible to show this line spatial gradient along AP axis through V1 (or maybe the center of V1 that the authors believe?), but at least to me the videos show more of the stagnant locations for both frequency bands as Figure 3 2D SVD results show. Also, although 3-6 Hz pattern is reasonably linear between PPA and V1 electrodes, the low gamma counterpart does not look like it. So nonlinear arguments here may not be appropriate, and just focus on the low gamma temporal evolution. However, the direction of the propagation from V1 to PPA is not really straight, so it is not clear if people would be convinced by the line propagation along AP axis through V1 channel.

Line 163: It is suggested to make sure to indicate “T” here is a transpose, and please add “,” after the equation. Then please eliminate the space (indentation) before where on line 164.

Line 188: Please change from “visual evoked” to “visually evoked”.

Lines 249-267: This is a very important point to be discussed to give us new insights on roles of slow and fast oscillations in the network dynamics especially in the context of the spatiotemporal dynamics.

Kopell's work compares beta and gamma oscillations, and it is purely based on the simulation studies, while other work in gamma oscillations (especially in the NHP work in the visual system) did not exclude the possibility of propagation of fast oscillations over space. Also, Supp Figure 4 shows that only a local phase gradient for V1 for the gamma oscillation, while V1 subfigure has almost a whole array depicted. It is extremely valuable to mention the mechanistic differences slow and fast wave patterns, but in the discussion there is a key citation missing in order to link V1 and PPA, Hovde K., et. Al. Eur J Neurosci. 2019;49:1313–1329, and please add this and other circuit/anatomical citations in the discussion section. Also, justification of not using layer annotations in PPA data needs to be given (in Figure 4).

Also, in comparison to Ermentrout and Kleinfeld 2001 paper which makes a claim that there are more synchronous activity dominates when a strong stimulus is present, as opposed to have observable wave propagation. What is the gap here?

Lines 295-311: Instead of looking at SFC at each site, it would be more meaningful to use SFC across sites, as supposed to just look at firing rates in V1 and PPA or only local SFC at each site. Also, it seems to be a laminar pattern for the spikes at least, and there are CSD plots in previous figures to Figure 7. The argument made between lines 306-311 is very informative to get insights on the neural signal processing, using the actual signals from both sites to characterize the signal propagations (and also compute for other pairs, to show that they are the actual hubs for both feedforward and feedback signal processing).

Line 315: Two types? "Two traveling waves" sound like two distinct waves are happening after visual stimuli. Please reword to clearly describe so that the meaning is not ambiguous.

Lines 364-366: It is not clear what this sentence means. Please elaborate and cite other work.

Line 376: Although it should be very uniform in terms of the impedances for the ECoG grids as well as the laminar probes, it would be informative to confirm this fact by putting the values in the manuscript.

Lines 380-382: Clearly there are conduction delays, but here the connections between V1 and PPA has more than one synaptic connection – both horizontally and interlaminarily. Please cite papers on anatomical studies illustrating underline circuits that are shown by now as well as even partial conduction speed characterizations to make sure that the wave speeds make sense physiologically. The following paragraph goes in deeper, but it would be insightful if authors can cite some anatomical and physiological work, as done in NHP motor cortical wave propagation of beta oscillations and supporting arguments for wave propagation speed.

Lines 404-405. It is indeed an attractive model to consider feedforward-feedback loop, but it is happening in real time. Thus, as mentioned above in relation to the heterogeneous slow LFPs (very consistent responses in the first two cycles and not afterwards) if there would be a speculation or a hypothesis as to what timing makes sense anatomically and physiologically how the visual stimulus is processed at which time scale and how long.

Lines 420-432: Citations here are very appropriate, but it is very curious to see a link of the current work to a recent work such as Wang T., et. Al, JNEUROSCI.1129-20.2020 to relate to the laminar subnetwork to superficial large scale spatiotemporal patterns.

Lines 617-625: It is not clear what makes it feasible to use the first 900 ms post stim durations to calculate phase-amplitude coupling. Also, this is computed for each area, V1 and PPA. If we are looking at the spatiotemporal patterns, what do these coupling indicate for the propagation patterns? It is not clear. Especially in V1 3-6 Hz responses, after two cycles of the oscillations, some portions of trials are flattened more. Figure 6d may be an attempt to show the clear relation between the local phase and amplitude, but it is not clear. Also, how about the timing at which the coupling happens Please articulate. Since the laminar profiles contain more information about the timing of potential propagations of the signals in addition to what ECoG signals mean in terms of the physiology, there should be more mechanistic interpretation of surface spatiotemporal patterns (and more citations are needed to look at the studies attempting to relate laminar profiles and surface recording (mostly by VSD)).

Line 644: Should it be Figure 7, as opposed to Figure 8?

Line 656: What is the justification to use 200 ms sliding window?

Lines 915-6: It is nice of authors to color-code the regions, but it would be nice to see each regional colors in stronger shades than the ones in the current manuscript.

Line 927: Although based on the response duration of V1 3-6 Hz (or maybe more like 3-10 Hz), is it fair to compute averages of ITPC over 800 ms for all the channels visualized in Figure 1g? Also based on this criterion to characterize the mean ITPC response across the array, there are several channels that indicate to have higher ITPC values for this duration at 3-6Hz range.

REVIEWER COMMENTS

We greatly appreciate the constructive critiques of our original manuscript and are grateful for the time, thought, and effort that the reviewers and the editor put into our work. We were very encouraged by the unanimous opinion of all reviewers that the work presented in our manuscript is interesting and important. Below we address every specific critique in turn. Here is a brief summary of the major additions of new data, analyses, and discussion.

- 1. Concern about non physiological stimuli (reviewer 1)** - we added new data illustrating that a new stimuli (static spatial gratings) also elicit feedforward and feedback waves (Sup. Figure 8). Using laminar recordings, we now show that in the PPA, 30-50 Hz oscillations exhibit intertrial phase coherence after a full screen flash (Sup. Figure 10, Sup Figure 11). This strongly implies that the feedforward wave reaches the PPA for weaker stimuli, but may not be adequately picked up at the dural surface. Finally, we show that after full screen flashes, unit activity is entrained by the slow feedback wave (Sup. Figure 14) and that the fraction of entrained units increases with increasing stimulus intensity in a similar way as the feedback wave. These data specifically address the concern that the waves are only observed with intense stimuli (LED flash).
- 2. Concern about presence of oscillations and filtering artifacts (reviewer 1)** - we added raw recordings of single trials (Sup. Figure 2) which clearly show both fast and slow oscillations. Furthermore, we show that after the stimulus, the spectra of single trial LFPs exhibit peaks in the 30-50 Hz and the 3-6 Hz range (Sup. Figure 2). Finally, we demonstrate that the results are not altered by the specific choice of filtering (Sup. Figure 6).
- 3. Questions about timing of the intertrial phase coherence (reviewer 3)** - We re-analyzed the ITPC using a different methodology (multitaper estimates) in non-overlapping temporal windows and show the time scale on which coherence evolves in newly added Sup. Figure 3. These results clearly demonstrate that the apparent increase in IPTC prior to the stimulus is an expected artifact of time-frequency uncertainty. We recomputed various phase correlation matrices using different windows. This did not alter our conclusions concerning the presence of interactions and better defined the dynamics of phase correlations.
- 4. Temporal dynamics of phase amplitude coupling (reviewer 3)** - As requested, we now show how phase amplitude coupling changes over time in newly added Sup. Figure 13.
- 5. Inter-site Spike Field Coherence (reviewer 3)** – As requested, we now show inter-site spike field coherence between V1 and PPA in newly added Sup. Figure 15 and discuss implications of these findings in the manuscript.
- 6. The potential role of feedback waves in sensory processing (reviewers 1-3)** - We now added a paragraph to the discussion that specifically addresses our hypothesis as to the role of these feedback waves.

We believe that as a result of addressing these critiques, the manuscript is much stronger. Thank you. Below, we specifically address every comment in detail. Please note that all figure callouts and line numbers we refer to in our responses correspond to the revised manuscript rather than the initial submission.

Reviewer #1 (Remarks to the Author):

In this work, the authors study ECoG and laminar recordings from awake mice following presentation of brief visual stimuli. By analyzing evoked activity patterns at both trial-average and single-trial level, they find the evoked response has a sophisticated spatiotemporal structure, with a narrowband feedforward wave in the 30-50 Hz range and a feedback wave in the 3-6 Hz range.

The results are timely and important; however, several major conceptual and technical points need to be addressed.

Thank you.

MAJOR

(1) A major concern is that the visual stimulus is far from physiologically realistic. This is not necessarily a detraction; however, the authors need to seriously consider what this means for their work. In standard visual paradigms, stimuli are either carefully constructed, titrated to produce specific visual responses, or matched to natural scenes. Further, it is a real possibility that the specific gamma activity and highly aligned spiking are result of the authors' specific and different stimulus presentation. The authors need to consider seriously this stimulus in the context of their work. While a strong, transient stimulus can certainly give much information about the system, it needs to be established that this information is general to how this system functions in other conditions, which already may not be the case with weaker visual stimuli (line 212-213). Additional experiments may be needed to clearly establish this because it's a serious concern.

Our response: Thank you for bringing up this important point. As you note, in this manuscript we are not studying how specific visual stimuli are processed or encoded in brain activity in any detailed way. Rather, our goal was to use a brief, simple stimulus in order to probe the structure of communication between different cortical areas involved in sensory perception. We chose to start with a brief, strong stimulus to consistently elicit LFP responses that could be recorded from the dural surface. Epidural LFP signals are greatly attenuated compared to intraparenchymal fields. Furthermore, epidural recordings are dominated by activity in the superficial cortical layers. This is especially relevant for detecting gamma oscillations that are much smaller in absolute amplitude relative to the slower oscillations. After characterizing our finding using the LED flash, we showed qualitatively similar results using lower luminance screen flashes (Sup. Figure 7).

As you requested, we added **additional new experiments** which characterized waves of activity evoked by presentation of a static spatial grating – a more standard stimulus used in vision research. We demonstrate (Sup. Figure 8) that these more conventional stimuli also elicit feedforward and feedback waves. We have also added new data (Sup. Figure 14) showing that in the immediate aftermath of lower luminance screen flashes, firing of individual neurons in V1 and PPA become correlated to the phase of the slow oscillation. Furthermore, the fraction of correlated neurons increases as a function of stimulus luminance in the same way as the extent of the feedback wave (Sup. Figure 14). Thus, it appears that the spatially extensive wave propagates in the feedback direction (see responses to comment 2 for more detail on definition and potential role of feedback) for both strong stimuli (LED flash) and more realistic stimuli (spatial gratings).

While the direction of the propagation of the feedforward wave was similar in all stimulation conditions, the extent was clearly curtailed for weaker (lower luminance) stimuli. There are several significant caveats, however, that need to be considered before concluding that the LED flash elicits a fundamentally

distinct feedforward wave pattern. Visually evoked gamma power is correlated to stimulus intensity in mice (Saleem et al, Neuron 2017). As the power of gamma oscillations is decreased, the signal to noise ratio of visual evoked gamma activity will also decrease. In an attempt to separate the evoked from spontaneous gamma oscillations, we only considered only the most visually evoked singular mode for phase analysis. However, as the amplitude of the evoked response (relative to background activity) decreases, SVD may not unequivocally identify the evoked component as a single mode. Thus, more sophisticated analysis may be required in the future to unequivocally characterize the spatial extent of the feedforward wave under weaker stimulation conditions. Newly added laminar recordings (Sup. Figure 10, Sup. Figure 11), however, show that intertrial gamma coherence is increased in the PPA with higher intensity screen flashes. This strongly argues that more conventional, weaker stimuli elicit spatiotemporal waves of activity that involve both the primary and the association cortices. Note that the peak of the intertrial phase coherence in the gamma range occurs below the surface of the cortex and may not therefore be adequately picked up epidurally. Future work using a high-density 3D recording configuration can address how the extent and features of the feedforward wave depend on specific features of the stimulus. We now explicitly acknowledge potential limitation of surface recordings for wave detection in the discussion (see also response to reviewer 3).

(2) The authors need to make a clear statement about what the feedback could be doing in this case. Top-down, feedback connections are largely thought to be anticipatory or modulatory. It's highly interesting that feedback activity could be organized as a wave; however, given the visual stimulus, it's highly unclear what feedback could be doing in this case. What is being anticipated or modulated in this experiment? This is a serious concern and one that must be addressed for this work to have impact in neuroscience. Finally, both the progression of activity across the visual system (as a feedforward "sweep") and the organization of feedforward-feedback dynamics across different frequency bands have been previously described in the work of Roelfsema. While specifically identifying these dynamics as a traveling wave is certainly interesting, the authors need to clearly describe how these results differ from that in previous work.

Our Response: Thank you for this insightful comment. The terms feedback and feedforward have somewhat different connotations depending on the context. In this work, we specifically focused on the anatomical direction of the propagation of the waves either from the primary visual cortex towards higher order association cortices (feedforward) or in the opposite direction (feedback) rather than on functional definition of top-down and bottom-up processing. This is similar to how feedforward and feedback processing has been described by Fries et al and Roelfsema et al and others. This feedforward and feedback designation also aligns with the laminar organization of these waves. As in previous work, the feedforward, gamma oscillations are initiated in L4 whereas feedback, slower oscillations involve layers 1-2 and 5.

While much previous work focused on the role of feedback processing in attention and other modulatory phenomena, there are reasons to believe that feedback signaling is a key component of sensory perception *per se*. It has been proposed by Baars, Varela, Lamme, and others that in order for the stimuli to gain conscious access, stimulus elicited activity must be coordinated across disparate cortical regions. The step-like increase in the extent of the feedback wave with increasing stimulus intensity (Sup. Figure 7) is consistent with its role in facilitating such coordination. Indeed, the behavior of the feedback wave is qualitatively similar to the "ignition" phenomena, required for conscious report, as described by (Bram van Vugt et al Science 2018). We now make this point explicit in the discussion of the manuscript. However, as we are not directly probing perception of the stimulus using a behavioral paradigm in this work, we would like to avoid making overly speculative claims.

It is true that Roelfsema and others identified that feedback and feedforward processing is associated with different temporal frequencies, and we cite their work extensively in the manuscript. Furthermore, the interactions between feedforward and feedback processing have been previously observed. These seminal findings in many ways inspired our work in the present manuscript.

There is, however, a fundamental distinction between the approach by Roelfsema and colleagues and our work. They focused on pairwise interactions between sites (e.g. V1 and V4). Observing a pairwise interaction between different sites does not, however, directly inform us about the overall macroscopic activity pattern. For instance, one could readily have feedback and feedforward interactions as in Roelfsema et al between V1 and V4 (e.g. PNAS 2014) and not have a clear wave like phenomena percolate across the cortex. One of the major implications of finding that both feedforward and feedback communication in the immediate aftermath of the stimulus forms traveling waves is that it allows for tremendous simplification of the spatiotemporal impulse response of the system. Future work should determine whether responses to sequences of stimuli (or stimuli presented in different locations in space) can be successfully understood as an interference pattern of spatiotemporal waves elicited by each stimulus in isolation. Furthermore, identification of spatiotemporal waves allows us to estimate macroscopic constants such as propagation velocities and spatial wavelengths (Sup. Figure 12). These, in turn arise from the biophysics of the networks that give rise to the waves. These macroscopic time and space constants will prove useful in constraining models of cortical networks and allow for direct interrogation of specific neurophysiological processes that give rise to the visually evoked traveling waves.

(3) It is important to consider potential distortions from narrowband filtering throughout this work. For both the 3-6 Hz and 30-50 Hz frequency band, it would be important to see examples of raw activity patterns, along with systematic quantification, to understand whether observed effects are due to true oscillations in each band. Importantly, activity near 50 Hz is known to contain spike waveform artifacts (Ray and Maunsell, PLoS Biology, 2008). Effects observed here in the higher frequency range extend up to about 80 Hz (cf. Fig 1d), clearly within the affected frequency range. Removing these artifacts, for example with the algorithm proposed by Zanos et al. (J Neurophys, 2011), has been observed to significantly impact spike-field relationships in this frequency range. Finally, the filtering approach of setting wavelet coefficients to zero (line 556-559) may not be optimal in this case, as it is almost reminiscent of a boxcar filter in the Fourier domain. The authors should establish that this is free from artifact and explain why this is a useful method.

Our response: This is indeed a very important point that we agonized over while writing the paper. To address your concern, we added more examples of raw traces (Sup. Figure 2) that show a clear oscillatory pattern evoked by the stimulus in both frequency ranges. For clarity, we plot raw LFP data on different time scales so that both fast and slow oscillations can be readily visualized. Also note that the average trace shown in Figure 1C exhibits a clear gamma ripple in the first ~100 ms after the stimulus. To quantitatively demonstrate that visual stimuli evoked relatively narrow band power in the LFP, we show the spectra of signals around the time of stimulus presentation in Sup. Figure 2. Note that in the immediate aftermath of the stimulus, there are clear peaks in the power spectra at the frequencies of interest. These observations are consistent with our major claim that the stimulus evokes distinct oscillations.

To make sure that our specific choice of filtering approach (wavelets) does not distort the data, we compared the results obtained with the wavelet-based filtering to those obtained with the conventional Butterworth filter (Figure below and Sup. Figure 6). To avoid phase distortions, the Butterworth filter was passed forward and backward across the data (implemented as *filtfilt* in Matlab). As can be seen from

Sup. Figure 6, the results are essentially identical regardless of the choice of filtering strategy. This is because unlike Fourier boxcar filters, Morlet wavelets are approximately Gaussian in the frequency domain. Given that our analysis of waves was based on epidural recordings (rather than in the brain parenchyma), we do not observe any spike waveform contamination in the filtered traces (see for instance single trial traces in the figure below or in the newly added Sup. Figure 2).

Thank you for pointing us towards algorithms aimed at removing spike artifacts from LFP filtered traces. As you note, these artifacts can lead to misleading spike field coherence estimates. However, Zanos et al show that the contamination of spike waveforms becomes relevant only when considering coupling between high frequency LFP oscillations (gamma range) and spikes. In contrast, in our manuscript, we are only considering spike-field coherence for slow (3-6 Hz) oscillations. As Zanos et al show in Figure 11, there was no difference between spike field coherence for the low frequency band (defined by them as < 24 Hz) before and after spike waveform removal. We have specifically examined the filtered waveforms (3-6Hz) and were not able to detect any evidence of a spike waveform (for instance traces shown below do not show any appreciable contributions from spike waveforms).

MINOR

- line 21: "It has been hypothesized...": By whom? Has this specific hypothesis of traveling waves been previously addressed?

Our Response: Thank you. We clarified our verbiage in the abstract.

- line 332: "Further, intracranial": ECoG recordings have certainly shown different patterns from early EEG; however, it may be incorrect to say they did not corroborate this earlier work.

Our Response: Thank you. We corrected the wording in the current manuscript.

- line 357: "numerical discrepancy": This word choice makes the difference seem like some sort of a mistake. Perhaps "quantitative difference" would be more clear.

Our Response: Thank you for pointing out our obtuse wording - we implemented your suggestion.

Reviewer #2 (Remarks to the Author):

This is a very interesting paper on visually evoked oscillatory activity in awake mice recorded using large ECoG grids (covering multiple areas in the mouse brain) providing a vast amount of novel analysis/ information on traveling waves in visual and association areas, and this within the known high frequency (gamma)/ feedforward and slow (theta) frequency/ feedback framework of signal processing. It is suggested that the demonstration of traveling waves in this context resolves an outstanding question in systems neuroscience: how the brain coordinates bidirectional network activity.

The paper is well written and the text and Figures guide well through the data. The results provide a lot of interesting information. The paper is in very good shape already. I don't have much suggestions for improvements/ additions.

Thank you very much.

Specific points (all discussion points)

1) I am convinced of the existence of traveling waves by the evidence presented. One thing that intrigues me though is how traveling waves can be reconciled with the concept of functional networks and connectivity. Functional and anatomical connectivity will in many instances skip cortical boundaries so that shortest paths for signal propagation will not always continuously follow the cortical surface, which is what traveling waves seem to largely do. Hence, there seems to be a fundamental difference between the function of signal propagation through anatomical connections versus along waves propagating across the cortical surface. I am unsure how to reconcile them. If waves travel across the cortex only when anatomical connections coincide with a more continuous connectivity pattern, then the term traveling waves I think is misleading, as whether waves are observed hinges on specific connectivity. If this is not the case, I am unsure how to integrate them in for instance signal propagation along larger fibre bundles that connect remote regions, e.g. the corpus callosum or the superior longitudinal fasciculus. Could this be discussed in more detail?

Our Response: The relationship between structural connectivity such as fiber bundles and functional connectivity (globally integrated activity patterns) is complex. A very important aspect of our work here is that essentially independent oscillations in neuronal activity at different cortical sites rapidly organize into a transient single macroscopic activity pattern that in many ways resembles a travelling wave. In other words, the waves observed after the stimulus, reflect functional synchronization between different local oscillations. As discussed by Kleinfeld and Ermentrout (Neuron 2001), travelling waves naturally emerge in networks where most connectivity is local. We believe that this local connectivity pattern in the cortex likely gives rise to the appearance of visually evoked travelling waves.

It is important to point out, as we did in the discussion and the results section, that the overall macroscopic pattern of activity deviates from that of canonical travelling waves. This can be seen by the curvature of the phase gradient vectors (Figure 3, Sup. Figure 7, Sup. Figure 8) and by the consequent spread in the estimates of the spatial wavelength and propagation velocity (Figure 5, Sup. Figure 12).

This deviation from the canonical traveling wave pattern can only be readily observed at macroscopic scales, such as the space covered by the ECoG grid. Along most small patches of the grid, the phase gradient is approximately linear. We suspect that this deviation from the canonical traveling wave pattern reflects the superimposition of short-range connectivity, which, in theory, gives rise to uniformly propagating waves, and long-range fiber bundles, which underlie structural connectivity. It is difficult at this point to provide firm evidence supporting this claim. However, our estimates of propagation velocities are roughly in the same range as the propagation velocity of cortico-cortical fibers. On the other hand, given the range of propagation velocities, other velocity estimates are more closely aligned with conduction in the white matter tracts and the reentrant cortico-thalamo-cortical fibers. These findings are consistent with the hypothesis that superimposition of multiple anatomic pathways which involve different time scales is ultimately responsible for the wave-like phenomena. Future work should further examine this issue with direct manipulation of specific anatomic pathways.

2) Another point that I find intriguing is that the slower (theta) oscillations occur much later in the trials than the faster (gamma) oscillation. How can this be reconciled with the prediction framework (referred to in the paper) where slower oscillations should appear before faster oscillations so that top-down control can occur? Could this be resolved?

Our response: This is indeed very interesting. Traditionally, feedback has been considered in the context of “top down” modulation such as attention, prediction, etc. But there are reasons to believe that feedback may be a component of sensory perception *per se*. Please see also our response to **Reviewer 1 Comment 2**. As we now address directly in the discussion, we hypothesize that the feedback wave may play a role in sensory perception. It has been shown that conscious perception of a salient stimulus is associated with large scale coordinated activity that can be recorded across the cortex. The feedback waves can provide a scaffold that coordinates this large-scale neuronal activity. It is really interesting to see how top-down influences like attention modulate the properties of this feedback wave. Future work should examine this question in detail.

Reviewer #3 (Remarks to the Author):

The current manuscript by Aggarwal A., et. al. characterizes the spatiotemporal patterns of visually evoked potentials across multiple frequency bands as travelling waves and propose such waves as a potential mechanism for feedforward-feedback visual (or potentially sensory in general) processing at the cortical level. Specifically, a large-scale simultaneous recording of cortical surface coupled with laminar recording from V1 and PPA was employed and showed that low gamma oscillation feedforward waves were initiated in V1 and propagated rostrally, while a rodent “alpha” oscillation over 3-6 Hz range feedback waves modulated the amplitude of the feedforward waves synchronizing spiking activities between V1 and PPA. The overall study design and the data are well done (except for the somewhat repeated stimuli that appeared to evoke responses prior to the stimulus onset), some of the analyses were not direct to show what the authors wanted to show. Furthermore, although at the very high level there was an intellectual motivation for the study, authors should include more citations for anatomical and physiological studies as opposed to cite more computationally driven studies. For the detailed comments, please see below.

Thank you.

Line 72: Most of this work -... > Most in this line of work ...

Our Response: Thank you. We corrected as you suggested.

Line 118: In Figure 1c, it is not clear how variable across trials the responses are during ~ 500-600 or 650 ms after the stim onset, and how the average peak is much smaller than the range of responses shown in individual gray traces... Not just the mean to be illustrated but adding std around the mean can be informative.

Our Comment: Thank you we now show the standard deviation of the trials in order to show the distribution about the mean LFP in a more quantitative fashion in Figure 1.

Lines 120-1: First of all, what is the statistical threshold for the significant ITPC based on the scale color-coded on the right of Figure 1d? Second, the ITPC values across 3-10Hz before the visual cue onset appear to be similar or in the same range as the “early” phase for the low gamma range. What is the logic here to say (or not to say) anything about the pre-stimulus response? Is this response a result of the semi-rhythmic stimuli (3-4 sec) so that animals could predict roughly when the stimulus appears? Also in other figures, V1 and PPA responses are compared, so why not showing the response of PPA channel in the same way as in Figure 1d?

Our Response: The threshold of statistical significance was $p < 0.05$. As now better described in the methods, ITPC for the real data was compared to time shifted surrogates. For each experimental preparation the experimentally observed ITPC estimate was expressed as a *z-score* (relative to time shuffled surrogates). To establish statistical significance for the entire dataset, a collection of such *z-scores* (one for each electrode in each preparation) was subjected to Stouffer's test. *P-value* threshold for statistical significance for each electrode was then adjusted using *Bonferroni* correction for multiple comparisons.

Increase in the ITPC (3-10Hz) before the stimulus is an artifact of time-frequency uncertainty. Wavelets of lower temporal scales cannot be precisely localized in time. We note that the stimuli were chosen from a uniform distribution on an interval between 3-4 seconds. Thus, it would not be possible for the mouse to anticipate the stimulus with precision of ~ 250 ms (which is the approximate interval by which the time-blurring artifact precedes the stimulus). To provide direct evidence for this artifactual smearing of the coherency band in time, we recomputed ITPC using multitaper spectral estimates in non-overlapping temporal windows. These newly added estimates show that, as expected, the coherency in both fast and slow frequency increases after the stimulus in Sup. Figure 3.

We have added an ITPC plot for the PPA channel to the figure as you requested in Figure 1 and Sup. Figure 3.

Lines 126-132: The early response, mentioned above in relation to Figure 1d, is prominent and visible in Figure 1i where V1 signals clearly have non-flat response compared to all the other three signals depicted in Figure 1 h and i. Also, it is not clear how readers are supposed to be convinced to have clear phase coupling from Figure 1i. Other measures such as cosine similarity or circular time series correlations to show statistically significant phase coupling in all the cases, especially in Figure 1i for the slower oscillation.

Our Response: We note that coherence (as defined in this manuscript) is a circular time series measure. To compute ITPC we only consider the phase of the complex-valued Morlet transform. Specifically, we convert all the complex outputs of the Morlet transform to vectors of unit length, where the angle of the vector is given by the arctan of the complex-valued wavelet coefficient. These unit vectors are then

averaged across trials at each time point in the evoked potential. The length of the resultant vector is shown by color in Figure 1g and 1h for fast and slow oscillations respectively. Thus, if phase angles were uniformly distributed on a circle, the length of the average vector would be zero. As we point out above the appearance of ITPC before the stimulus, especially prominent for slow oscillations, is an artifact. Time course of ITPC is independently estimated in Sup. Figure 3. However, as you correctly point out ITPC (as described above) does not directly speak to the consistency of phase offset between electrodes. While these inter channel relationships are reflected in the SVD, to address your concern directly, we added a phase difference “rose” plot to Figure 1k-l to illustrate that indeed V1 and PPA LFP oscillations have a consistent nonzero phase offset at both 30-50 Hz and 3-6 Hz frequencies.

Lines 133-149: The authors' intent is somewhat visible to show this line spatial gradient along AP axis through V1 (or maybe the center of V1 that the authors believe?), but at least to me the videos show more of the stagnant locations for both frequency bands as Figure 3 2D SVD results show. Also, although 3-6 Hz pattern is reasonably linear between PPA and V1 electrodes, the low gamma counterpart does not look like it. So nonlinear arguments here may not be appropriate, and just focus on the low gamma temporal evolution. However, the direction of the propagation from V1 to PPA is not really straight, so it is not clear if people would be convinced by the line propagation along AP axis through V1 channel.

Our Response: You are indeed correct. The movies show a more complex pattern of wave propagation that is not exactly the same as would be expected for a prototypical travelling wave. We discuss this observation in the results section (results on spatial wavelength and propagation speed) and in the discussion. See also our response to Reviewer 2. There is however no discrepancy between the data in Figure 2, movies, and the SVD results. There is an important difference in what is being shown in these figures/movies. In Figure 2 we show just the average of the traces filtered at the frequency of interest. The same kind of data are shown in the movie but on the entire 2D grid of electrodes. However, as we point out in the results, this trial averaging does not directly reveal wave propagation on a single trial level or necessarily isolates just the stimulus evoked activity. This is the reason why we undertook the SVD analysis of single trials and the separation of the visually evoked modes. Note that the vertical component of the SVD phase gradient vectors (at ML coordinate of -2.25) predominantly points up for the gamma wave and down for the 3-6Hz wave. This is why, when looking just along a column of electrodes, the wave appears to propagate just along the AP axis in the feedforward direction. However, as the SVD results show and as we discuss, the overall pattern of wave propagation is curved. This curvature (or deviation from linearity) is what gives rise to a distribution of wavelengths and propagation speeds at different spatial scales and locations in the cortex (Figure 5, Sup. Figure 12). The curvature of this path is what gives the movies a more complex appearance that has some characteristics in common with travelling waves in a passive medium (still water) but also exhibit some important differences from this simple example of a wave. Please see responses to **Reviewer 2 Comment 1** discussing the implications of this deviation from linearity.

Line 163: It is suggested to make sure to indicate “T” here is a transpose, and please add “,” after the equation. Then please eliminate the space (indentation) before where on line 164.

Our Response: Thank you. We clarified the equation as you suggested.

Line 188: Please change from “visual evoked” to “visually evoked”.

Our Response: Thank you. We corrected this in the manuscript.

Lines 249-267: This is a very important point to be discussed to give us new insights on roles of slow and fast oscillations in the network dynamics especially in the context of the spatiotemporal dynamics. Kopell's work compares beta and gamma oscillations, and it is purely based on the simulation studies, while other work in gamma oscillations (especially in the NHP work in the visual system) did not exclude the possibility of propagation of fast oscillations over space. Also, Supp Figure 4 shows that only a local phase gradient for V1 for the gamma oscillation, while V1 subfigure has almost a whole array depicted. It is extremely valuable to mention the mechanistic differences slow and fast wave patterns, but in the discussion there is a key citation missing in order to link V1 and PPA, Hovde K., et. Al. Eur J Neurosci. 2019;49:1313–1329, and please add this and other circuit/anatomical citations in the discussion section. Also, justification of not using layer annotations in PPA data needs to be given (in Figure 4).

Our Response: We agree that others have conjectured that the phase coupling between different oscillations gives rise to a travelling wave. Indeed, as we discussed in the paper, travelling waves have been observed on a more local scale. Our major contribution here is that these traveling waves involve much of the cortex, are separated in frequency, and propagate in feedforward and feedback directions on a large scale. Thus, there is no conflict between our work here and previous findings examining pairwise phase relationships between signals. Please see our response to **Reviewer 1 Comment 1** on the spatial extent of the gamma wave with different stimulation conditions. Also, please see newly added data on spike synchronization in the PPA and V1 with different stimuli (Figure 14). Finally, we added a different stimulation condition (spatial grating) (Sup. Figure 8). Under all stimulation conditions we observe both the feedback and the feedforward wave.

Thank you for pointing out some important work on connectivity between V1 and PPA. We now cite it in the manuscript. In the manuscript, we defined V1 layers neurophysiologically on the basis of responses to visual stimuli. As responses of the PPA to visual stimuli in mice have not been previously well established, we did not feel that we could make any layer claims in the PPA unambiguously. However, as you requested in our analysis of inter-site SFC (Sup. Figure 15), we do approximately assign the units in PPA to different layers. Throughout the paper, we do provide our best estimates of the cortical depth at which each signal was recorded in the PPA. These can be approximately aligned to the layers in the PPA using the anatomical information that we now cite in the manuscript.

It is difficult to directly relate the modeling work of Kleinfeld and Ermentrout to the experimental findings in the manuscript. We observe wave propagation with both strong and weak stimuli. However, as we point out in responses to other comments, these phenomena are not purely traveling waves. The nonlinearities in the phase gradient strongly suggest that these phenomena arise as a consequence of superimposition of signaling along multiple neuroanatomical pathways with different conduction delays and connectivity patterns. These inhomogeneities were not considered in the highly simplified models of Kleinfeld and Ermentrout.

Also, in comparison to Ermentrout and Kleinfeld 2001 paper which makes a claim that there are more synchronous activity dominates when a strong stimulus is present, as opposed to have observable wave propagation. What is the gap here?

Our Response: Please see above for the discussion of the work of Kleinfeld and Ermentrout. We believe that this deviation from the predictions made on the basis of simplified models, may suggest that the models need to be refined to incorporate more anatomical details of multiple pathways that likely contribute to the wave-like phenomena observed experimentally.

Lines 295-311: Instead of looking at SFC at each site, it would be more meaningful to use SFC across sites, as supposed to just look at firing rates in V1 and PPA or only local SFC at each site. Also, it seems to be a laminar pattern for the spikes at least, and there are CSD plots in previous figures to Figure 7. The argument made between lines 306-311 is very informative to get insights on the neural signal processing, using the actual signals from both sites to characterize the signal propagations (and also compute for other pairs, to show that they are the actual hubs for both feedforward and feedback signal processing).

Our Response: Thank you. These data are shown in newly added Sup. Figure 15.

Line 315: Two types? “Two traveling waves” sound like two distinct waves are happening after visual stimuli. Please reword to clearly describe so that the meaning is not ambiguous.

Our Response: We clarified this issue, thank you. At multiple points in the manuscript and in several figures, we show that the two waves interact (phase amplitude coupling and spike field coherence). Thus, the two waves are best thought of as two aspects of a single macroscopic dynamic elicited by the stimulus.

Lines 364-366: It is not clear what this sentence means. Please elaborate and cite other work.

Our Response: Thank you for pointing out the confusion. We clarified our verbiage in the abstract.

Line 376: Although it should be very uniform in terms of the impedances for the ECoG grids as well as the laminar probes, it would be informative to confirm this fact by putting the values in the manuscript.

Our Response: We added the impedances for the ECoG grids and laminar probes into the methods as you suggested.

Lines 380-382: Clearly there are conduction delays, but here the connections between V1 and PPA has more than one synaptic connection – both horizontally and interlaminarily. Please cite papers on anatomical studies illustrating underline circuits that are shown by now as well as even partial conduction speed characterizations to make sure that the wave speeds make sense physiologically. The following paragraph goes in deeper, but it would be insightful if authors can cite some anatomical and physiological work, as done in NHP motor cortical wave propagation of beta oscillations and supporting arguments for wave propagation speed.

Our Response: Thank you. We have now added more citation on the anatomical connections between V1 and PPA and relate these anatomical findings to our experimental observations in the discussion.

Lines 404-405. It is indeed an attractive model to consider feedforward-feedback loop, but it is happening in real time. Thus, as mentioned above in relation to the heterogeneous slow LFPs (very consistent responses in the first two cycles and not afterwards) if there would be a speculation or a hypothesis as to what timing makes sense anatomically and physiologically how the visual stimulus is processed at which time scale and how long.

Our Response: We have characterized the dynamics of the rise and fall of the ITPC in more detail (Sup. Figure 3) as you requested (see above),. We have added a paragraph to the discussion that addresses the potential role of feedback waves. We would rather not make any overly speculative claims about the mechanisms that give rise to these dynamics as these mechanisms are not directly addressed in the paper.

Lines 420-432: Citations here are very appropriate, but it is very curious to see a link of the current work to a recent work such as Wang T., et. Al, JNEUROSCI.1129-20.2020 to relate to the laminar subnetwork to superficial large scale spatiotemporal patterns.

Our Response: Thank you for drawing our attention to this important work. We added a reference to Wang et al, 2020 in our revision.

Lines 617-625: It is not clear what makes it feasible to use the first 900 ms post stim durations to calculate phase-amplitude coupling. Also, this is computed for each area, V1 and PPA. If we are looking at the spatiotemporal patterns, what do these coupling indicate for the propagation patterns? It is not clear. Especially in V1 3-6 Hz responses, after two cycles of the oscillations, some portions of trials are flattened more. Figure 6d may be an attempt to show the clear relation between the local phase and amplitude, but it is not clear. Also, how about the timing at which the coupling happens. Please articulate. Since the laminar profiles contain more information about the timing of potential propagations of the signals in addition to what ECoG signals mean in terms of the physiology, there should be more mechanistic interpretation of surface spatiotemporal patterns (and more citations are needed to look at the studies attempting to relate laminar profiles and surface recording (mostly by VSD)).

Our Response: Thank you for highlighting this very important issue. As you suggested, we now shortened the window in which we calculate phase amplitude coupling. As you point out, the strength of the interaction does indeed depend on the time and peaks in the first 450 ms after the stimulus. To better define the time course of the phase amplitude coupling, we now computed phase amplitude coupling using a sliding 450 ms window. These results are shown in the newly added Sup. Figure 13. We have also added some references to the complexities involved in relating VSD and ECoG recordings to specific neurophysiological processes in the discussion. As you know this relationship is highly nontrivial, and full exposé of this issue is beyond the scope of this paper. Nevertheless, we completely agree with you that some discussion of this was merited.

Line 644: Should it be Figure 7, as opposed to Figure 8?

Our Response: Thank you. We corrected the label.

Line 656: What is the justification to use 200 ms sliding window?

Our Response: The firing rate of V1 and PPA neurons is fairly low (<10 Hz). Thus, using smaller windows would result in highly noisy firing rate estimates. These firing rate estimates are used to construct the null (Poison) models of neuronal firing which are used as a comparison to real recordings. We note that the window was shifted by 1 ms to better approximate the time course of rate modulation. We explored some different choices of the window and did not detect any significant differences in the conclusions.

Lines 915-6: It is nice of authors to color-code the regions, but it would be nice to see each regional colors in stronger shades than the ones in the current manuscript.

Our Response: Thank you. We increased the saturation of the colored regions as you suggested.

Line 927: Although based on the response duration of V1 3-6 Hz (or maybe more like 3-10 Hz), is it fair to compute averages of ITPC over 800 ms for all the channels visualized in Figure 1g? Also based on this

criterion to characterize the mean ITPC response across the array, there are several channels that indicate to have higher ITPC values for this duration at 3-6Hz range.

Our Response: Thank you for pointing out the effect that timing might have on our ITPC results. We chose 800 ms for the time window to average the 3-6 Hz wavelet based ITPC over each channel location in the grid to account for the notion that ITPC might increase at different time points in a location specific fashion. In order to better describe the temporal dynamics of ITPC over the cortical surface, we computed ITPC using multitaper spectral analysis over non-overlapping windows and averaged the ITPC within each window across channels (Sup. Figure 3e).

REVIEWER COMMENTS

Reviewer #1 (Remarks to the Author):

In this revision, the authors have adequately addressed all concerns raised in the first round of review.

The only point to be addressed prior to publication is on line 349: "Further, intracranial recordings (ECoG) exhibited different spatiotemporal patterns compared to those discovered in EEG recordings (Bahramisharif et al., 2013; Hangya et al., 2011; Mak-McCully et al., 2015; Muller et al., 2018), leaving the existence and functional role of macroscopic traveling waves in question." This is incorrect: this sentence states that ECoG recordings exhibit different spatiotemporal patterns than those in EEG recordings, and then goes on to state that this difference leaves the existence and functional role of macroscopic traveling waves unclear. However, it is well known that ECoG recordings are a more direct reflection of cortical activity than EEG, and several studies (as cited) have reported traveling waves in ECoG. With this in mind, it may be misleading to suggest the difference between spatiotemporal dynamics in these two recording types leaves the existence of macroscopic traveling waves unclear. This point should be corrected prior to publication.

Reviewer #2 (Remarks to the Author):

All my comments (minor) to what was already in the previous version a very interesting paper that provides a vast amount of novel analysis/information have been addressed. I have no further comments

Reviewer #3 (Remarks to the Author):

Thank you for item-by-item responses and thoughtfully addressing issues raised by the reviewers.

There are some more issues to be addressed to move to the next stage.

From the previous comment:

Our Comment: Thank you we now show the standard deviation of the trials in order to show the distribution about the mean LFP in a more quantitative fashion in Figure 1.

☐ Thank you for adding STDs around the mean. But especially compared with the figure in the previous version, some of the trial variations (or maybe there exists a few types of responses) blurred out. Can you at least comment on that point? Especially now that specific relation of phase to LFPs was investigated in Figure 7 and strength of spike-LFP entrainment can change after a few cycles in V1 (and less significantly in PPA) as shown in Figure 7.c and 7.d, how about single trial waveforms?

Additional comments:

L694-: Although the title of the section is “Velocity of visual evoked waves:”, actual definition of the wave velocity is not described here. Please make sure to describe the method to calculate the wave velocity so that readers can understand.

Especially regarding to wave characterizations presented in Figure 3, heterogeneous or non-planer waves are observed (although the current manuscript nicely focused on the relation between V1 and PPA). Thus, a recent work such as

Traveling waves in the prefrontal cortex during working memory, by Sayak Bhattacharya, Scott L. Brincat, Mikael Lundqvist, Earl K. Miller. <https://doi.org/10.1371/journal.pcbi.1009827>

and references therein especially in terms of the non-planer wave characterization would be nice to be included in the citations and potential implications of functions of travelling waves in the neural processing (or sensory processing specifically) to be discussed in the Discussion.

Addition of Figure 7 is wonderful, but given that each areal spike phase characterization is done based on layers/depth, why all the pairs are mixed together in Figure 7.g and 7.h? Any information about phase for units and its relation to correlational spiking across PPA and V1?

Reviewer Comments

We would like to thank the reviewers for their constructive criticism of the revised manuscript. We were very glad to see that all reviewers felt that we have addressed all of the significant critiques of the original manuscript. In this, second revision, we address the minor concerns that remain. Thank you again for your time and suggestions.

Reviewer #1 (Remarks to the Author):

In this revision, the authors have adequately addressed all concerns raised in the first round of review.

Thank you very much.

The only point to be addressed prior to publication is on line 349: "Further, intracranial recordings (ECoG) exhibited different spatiotemporal patterns compared to those discovered in EEG recordings (Bahramisharif et al., 2013; Hangya et al., 2011; Mak-McCully et al., 2015; Muller et al., 2018), leaving the existence and functional role of macroscopic traveling waves in question." This is incorrect: this sentence states that ECoG recordings exhibit different spatiotemporal patterns than those in EEG recordings, and then goes on to state that this difference leaves the existence and functional role of macroscopic traveling waves unclear. However, it is well known that ECoG recordings are a more direct reflection of cortical activity than EEG, and several studies (as cited) have reported traveling waves in ECoG. With this in mind, it may be misleading to suggest the difference between spatiotemporal dynamics in these two recording types leaves the existence of macroscopic traveling waves unclear. This point should be corrected prior to publication.

We agree that this sentence did not come across as we intended and modified it. The present form reads as follows: "Further, intracranial recordings (ECoG) that suffer from fewer signal distortions than the EEG, exhibited different spatiotemporal patterns compared to those discovered in EEG recordings (Bahramisharif et al., 2013; Hangya et al., 2011; Mak-McCully et al., 2015; Muller et al., 2018). Novel experimental imaging techniques using voltage sensitive dyes (VSDs) reveal mesoscopic travelling waves that are confined by anatomical boundaries between cortical regions (Muller et al., 2014; Polack and Contreras, 2012) or produce complex interference patterns at the inter-region boundaries (Xu et al., 2007)."

Reviewer #2 (Remarks to the Author):

All my comments (minor) to what was already in the previous version a very interesting paper that provides a vast amount of novel analysis/information have been addressed. I have no further comments.

Thank you very much.

Reviewer #3 (Remarks to the Author):

Thank you for item-by-item responses and thoughtfully addressing issues raised by the reviewers.

Thank you very much.

There are some more issues to be addressed to move to the next stage.

From the previous comment:

Our Comment: Thank you we now show the standard deviation of the trials in order to show the distribution about the mean LFP in a more quantitative fashion in Figure 1.

Thank you for adding STDs around the mean. But especially compared with the figure in the previous version, some of the trial variations (or maybe there exists a few types of responses) blurred out. Can you at least comment on that point? Especially now that specific relation of phase to LFPs was investigated in Figure 7 and strength of spike-LFP entrainment can change after a few cycles in VI (and less significantly in PPA) as shown in Figure 7.c and 7.d, how about single trial waveforms?

We realize that the plots of the superimposed single trials are not very conducive to the inspection of single trials. In this revision, we added (as supplementary figure 2A-B), a plot where ~ 100 single trials are shown as a heatmap. We believe this makes it easier to examine the voltage waveforms on a single trial basis. As is well known, there is some variability between trials. The issue of existence of distinct types of responses, however, is a very complex one and is well beyond the scope of this paper. In preliminary analysis we attempted to separate single trial responses into distinct types but were not able to do so reliably. However, to make any claims about the existence (or lack thereof) of distinct types of responses would require a much more in-depth investigation. In this manuscript, we focus on the feedforward and feedback waves that are reliably observed on a single trial basis.

Additional comments:

L694-: Although the title of the section is “Velocity of visual evoked waves:”, actual definition of the wave velocity is not described here. Please make sure to describe the method to calculate the wave velocity so that readers can understand.

We agree that the explanation of wave velocity calculation was not adequately fleshed out in the original submission. We have now modified this description to add the actual equation used. Here, we paste the newly added text describing the definition of wave velocity: The instantaneous temporal frequency was computed by measuring the slope of the unwrapped temporal phase of the most visually responsive mode. This yields the temporal frequency $f_t = \frac{d\theta_t}{dt}$. The spatial gradient at each location (x,y) is a two component vector $\left\langle \frac{d\theta_s}{dx}, \frac{d\theta_s}{dy} \right\rangle_{(x,y)}$, where dx and dy are unit vectors in the mediolateral and the anterior-posterior directions respectively. The

Euclidian norm of this vector is $\nabla_s = \sqrt{\left(\frac{d\theta_s}{dx}\right)^2 + \left(\frac{d\theta_s}{dy}\right)^2}$. The wave velocity at each location is then defined as $v = \frac{f_t}{\nabla_s}$.

Especially regarding to wave characterizations presented in Figure 3, heterogeneous or non-planer waves are observed (although the current manuscript nicely focused on the relation between V1 and PPA). Thus, a recent work such as

Traveling waves in the prefrontal cortex during working memory, by Sayak Bhattacharya, Scott L. Brincat, Mikael Lundqvist, Earl K. Miller. <https://doi.org/10.1371/journal.pcbi.1009827> and references therein especially in terms of the non-planer wave characterization would be nice to be included in the citations and potential implications of functions of travelling waves in the neural processing (or sensory processing specifically) to be discussed in the Discussion.

Thank you for pointing us to this newly published study with direct relevance to our work. We have added the reference to it as you suggested.

Addition of Figure 7 is wonderful, but given that each areal spike phase characterization is done based on layers/depth, why all the pairs are mixed together in Figure 7.g and 7.h? Any information about phase for units and its relation to correlational spiking across PPA and V1?

As requested, we now sort the neuronal pairs by location of the V1 and the PPA spike and modify Figure 7 accordingly.

REVIEWERS' COMMENTS

Reviewer #3 (Remarks to the Author):

Thank you again for thorough responses to each items raised in the last version of the manuscript. I am happy to recommend to provisionally accept the current manuscript for publication. Congratulation!

REVIEWERS' COMMENTS: Round 3

Reviewer #3 (Remarks to the Author):

Thank you again for thorough responses to each items raised in the last version of the manuscript. I am happy to recommend to provisionally accept the current manuscript for publication. Congratulation!

Thank you so much for your revisions. We are confident that these revisions have made the paper stronger.